# Generating Novel Scene Compositions from Single Images and Videos

## Abstract

Given a large dataset for training, GANs can achieve remarkable performance for the image synthesis task. However, training GANs in extremely low data regimes remains a challenge, as overfitting often occurs, leading to memorization or training divergence. In this work, we introduce SIV-GAN, an unconditional generative model that can generate new scene compositions from a single training image or a single video clip. We propose a two-branch discriminator architecture, with content and layout branches designed to judge internal content and scene layout realism separately from each other. This discriminator design enables synthesis of visually plausible, novel compositions of a scene, with varying content and layout, while preserving the context of the original sample. Compared to previous single-image GANs, our model generates more diverse, higher quality images, while not being restricted to a single image setting. We show that SIV-GAN successfully deals with a new challenging task of learning from a single video, for which prior GAN models fail to achieve synthesis of both high quality and diversity.

Training video and samples generated from *a single video*

Training image and samples generated from *a single image*

Figure 1: Images generated by SIV-GAN. Our model successfully operates in extremely low data regimes, generating new scene compositions with varying content and layout from a single video (first two rows) or a single image (last three rows). For example, from the single training video with a car on the road, SIV-GAN generates images without a car or with two cars; and for the single surfing image, it can synthesize layouts with a different position and configuration of waves varying the number of surfers in the scene. SIV-GAN is able to maintain the quality and diversity of scene compositions while operating at different image resolutions, e.g., 192×320 (fourth row) and 512×896 (last row). (Original training samples are shown in grey or red frames.)

# 1 INTRODUCTION

The quality of synthetic images produced by generative adversarial networks (GANs) (Goodfellow et al., 2014) has greatly improved in recent years (Zhang et al., 2019; Brock et al., 2019; Park et al., 2019; Karras et al., 2019; Lin et al., 2019; Schönfeld et al., 2020; Karras et al., 2020b). These impressive results are in large part enabled by the availability of large, diverse datasets, typically consisting of tens of thousands of images. This dependency on the availability of training data limits the applicability of GANs in domains where collecting a large dataset is not feasible. In some real-world applications, collection of even a small dataset remains challenging due to specific constraints related to privacy, copyright status, subject type, geographical location, time, and dangerous or hazardous environments. It may happen that rare objects or events are present only in one image or video, and it is difficult to obtain a second one. For example, this includes images of exclusive artworks or videos of traffic accidents recorded in extreme conditions. Enabling GANs to learn in the extremely low data regimes, such as learning from the single data instance examples, thus has the potential to improve their utilization in practice.

Prior work (Shocher et al., 2019; Shaham et al., 2019; Hinz et al., 2021) focused only on learning generative models from a *single image* (Fig. 1, rows 3-5). In this work, we introduce a new task of learning to synthesize images from a *single video* (Fig. 1, rows 1-2), where the training data is a collection of frames from a 2-10 second video clip. In practice, capturing such a short video takes almost as little effort as collecting a single image. However, compared to a single image, a video contains much more information about the scene and the objects of interest (e.g., different poses and locations of objects, various camera views, dynamic backgrounds). Learning from a video can enable synthesis of higher quality, more diverse images, while still keeping very low requirements on data availability, and therefore can improve GAN usability for practical applications.

Training GANs in such extremely low data regimes is challenging as the models are prone to overfitting, leading to replication of training data or poor synthesis quality due to training instability (Karras et al., 2020a). For example, applying the few-shot image synthesis models, such as FastGAN (Liu et al., 2021), to learn from a single image or video leads to severe memorization problems (see Sec. 4). To mitigate memorization, single-image GAN models (Shaham et al., 2019; Hinz et al., 2021) proposed to learn an internal patch-based distribution of an image, employing a cascade of multi-scale patch-GANs (Isola et al., 2017) trained in multiple stages. Though these models overcome memorization, producing different versions of a training image, they cannot learn high-level semantic properties of the scene, e.g., to dissect objects from the background. They often suffer from incoherent shuffling of image patches, distorting objects and producing unrealistic layouts (see Fig. 4 and 5). Furthermore, they have more difficulty to generate realistic scenes from a single video.

We aim to go beyond patch-based learning, seeking to generate novel, plausible compositions of objects in the scene, while preserving the original image context. We want to learn a model which is able to compose new layouts, re-arrange objects in the scene, remove or duplicate instances, and change their shape and size. The new scene compositions should be visually plausible, with objects preserving their appearance and natural shape, and scene layouts looking realistic to the human eye. Reaching such synthesis quality from a single image or a single video is notably challenging, as the model needs to learn to distinguish objects and background using only one image or few training frames, without any direct object supervision (e.g., mask annotations).

To this end, we introduce SIV-GAN (**S**ingle **I**mage and **V**ideo GAN), an unconditional, one-stage GAN model, capable of learning from the single data instances to generate images that are substantially different from the original training sample, while still preserving its context. This is achieved by two key ingredients: the novel discriminator design and the proposed diversity regularization for the generator. Our discriminator has two branches, separately judging image content and scene layout realism. The content branch evaluates objects' fidelity irrespective of their spatial arrangement, while the layout branch looks only at the global scene coherency. Disentangling the discriminator's decision about content and layout helps to prevent overfitting and provides a more informative signal to the generator. To achieve a high diversity among generated samples, we further extend the regularization technique of (Yang et al., 2019) to unconditional image synthesis in single data instance regimes. The prior work of (Yang et al., 2019) encourages the generation of different images depending on their input latent codes, thus the difference between images is proportional to the distance between their codes in the latent space. Assuming that in case of a single image or a single video all generated images should belong to one semantic domain (i.e. preserve the original training sample context), and thus should be more or less equally different from each other, we apply

diversity regularization uniformly, independent of the latent space distance. Moreover, we use the regularization in the feature space, inducing both high- and low-level diversity.

We demonstrate the effectiveness of our model for the single image setting as well as for the novel single video setting in Sec. 4. SIV-GAN is the first model that successfully learns in both of these extremely low-data settings, improving over prior work (Shaham et al., 2019; Hinz et al., 2021; Liu et al., 2021) in both image quality and diversity. In contrast to FastGAN (Liu et al., 2021), our model does not suffer from memorization (see Table 1 and 2); while compared to single-image GANs (Shaham et al., 2019; Hinz et al., 2021), our model does not distort objects, preserving their appearance. In summary, our contributions are: 1) We propose a new task of learning generative models from a single video. 2) We present a novel two-branch discriminator, encouraging the generation of new scene compositions with layouts and content substantially different from training samples. Our proposed diversity regularization ensures a high variability among generated samples in the challenging single data instance regimes. 3) With SIV-GAN, we achieve high quality and diversity when learning from both single images and videos, outperforming prior work on two different datasets.

## 2 RELATED WORK

**Single Image Generative Models.** Several works have investigated learning generative models from a single image. Ulyanov et al. (2018); Shocher et al. (2018) showed that when trained on a single image, a deep convolutional network can learn a useful representation that captures the internal statistics of that image. These learned representations can be employed to synthesize textures from a sample texture image (Jetchev et al., 2016; Bergmann et al., 2017; Ulyanov et al., 2016; Zhou et al., 2018; Li & Wand, 2016; Ulyanov et al., 2017), to "blindly" super-resolve (Shocher et al., 2018; Bell-Kligler et al., 2019; Lugmayr et al., 2020), or to inpaint the image (Ulyanov et al., 2018). Recently, single image GAN models (Shocher et al., 2019; Shaham et al., 2019) have been proposed, revealing the power of image priors learned from a single natural image for synthesis and manipulation tasks. InGAN (Shocher et al., 2019) introduced a GAN model conditioned on a single natural image, which can remap the input to any size or shape while preserving its internal patch-based distribution. SinGAN (Shaham et al., 2019) employed an unconditional GAN to produce images of arbitrary size from noise, and used a multi-stage training scheme to learn the multi-scale patch distribution of an image. ConSinGAN (Hinz et al., 2021) extended SinGAN by improving the rescaling for multi-stage training and training several stages concurrently, which enabled reducing the size of the model and making the training more efficient. Alternatively, instead of using adversarial training Granot et al. (2021) proposed to use a patch-nearest-neighbour search.

Contrary to single image GANs (Shaham et al., 2019; Hinz et al., 2021), our model is trained in a single stage, and is designed to learn not only the internal patch-based distribution of an image, but also to capture high-level content, such as scene layouts or appearance of objects. The latter enables more diverse, higher quality synthesis (see Table 1). Our model is also more universal: it achieves high quality and diversity when learning from both single images and videos, and thus can be applied to a wider range of real-world scenarios.

**Few-Shot Generative Models.** Few-shot image generation seeks to generate more data of a given domain, with only a few training samples available. Many existing methods resorted to using large image sets of seen classes to train a conditional GAN, which is then used to generate new images for an unseen class based on a few examples (Hong et al., 2020a; Antoniou et al., 2017; Hong et al., 2020b). Others focused on improving the sample efficiency of GANs by few-shot adaptation. These methods usually start from GANs pre-trained on large datasets, and adapt them on a few samples in the target domain by fine-tuning the generator and discriminator weights (Wang et al., 2018; Mo et al., 2020), changing batch statistics (Noguchi & Harada, 2019), restricting the space of trainable parameters (Robb et al., 2020), transferring the knowledge via a miner network (Wang et al., 2020), or enforcing cross-domain consistency (Ojha et al., 2021). Unlike our work, the above methods heavily rely on external knowledge from a pre-trained GAN. Thus, their performance highly depends on the semantic consistency between the target domain and the pre-trained model, and their incompatibility can lead to even worse results (Zhao et al., 2020a).

Most recently, FastGAN (Liu et al., 2021) proposed to train an unconditional GAN for few-shot image synthesis from scratch. To avoid overfitting, they enabled a fast learning of the generator through a skip-layer channel-wise excitation module, and used a self-supervised discriminator to continuously provide useful signals to the generator. However, FastGAN still suffers to successfully

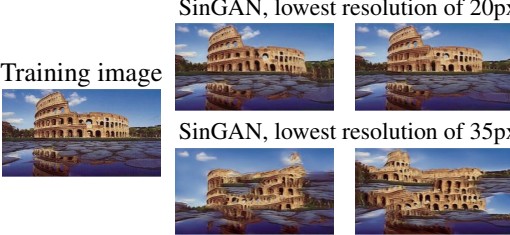

Figure 2: SIV-GAN. The two-branch discriminator judges content separately from the scene layout realism and thus enables the generator to produce images with varying content and global layouts.

learn from a single image or video. Although the single video setting has similar number of training frames to the few-shot setting (∼100 samples), it provides much less variability in training data due to high correlation of adjacent frames, leading FastGAN to memorization issues (see Fig. 5).

Another line of work focused on improving the stability of GANs when trained with limited data. To prevent the discriminator from overfitting, Karras et al. (2020a); Zhao et al. (2020b;a) explored differentiable data augmentations (DA) for both real and generated images. Tseng et al. (2021) proposed to regularize GANs under limited data with a LeCam consistency loss, showing it to be complementary to DA techniques. These works used limited, but still relatively large training sets ($\geqslant$ 1k images) compared to the few-shot and single video regimes ($\leqslant$ 100). As shown in Sec. 4.2, using DA alone is not enough to achieve good diversity when learning from a single image or video.

## 3   SIV-GAN

In this section, we present SIV-GAN, an unconditional GAN model that learns from a single image or a single video to generate new plausible compositions of a given scene with varying content and layout. The key ingredients of SIV-GAN are a novel design of a two-branch discriminator (Sec. 3.1) and a diversity regularization introduced for synthesis in single data instance regimes (Sec. 3.2).

### 3.1   CONTENT-LAYOUT DISCRIMINATOR

One challenge of training GANs in single data instance regimes is the problem of overfitting to original samples. In many cases the model can simply memorize the original training images and their augmented versions (if used during the training). To avoid this memorization effect, Shaham et al. (2019) and Hinz et al. (2021) proposed to learn an internal patch-based distribution of a single image by using a hierarchy of patch-GANs (Isola et al., 2017; Li & Wand, 2016) at different image scales. As the employed patch-GANs have small receptive fields and limited capacity, they are prevented from memorizing the full image. However, the downside of training each scale of the patch-GANs in a separate stage is that any layout decisions made by the coarser scale generators cannot be corrected at later, finer generation stages. Thus, both the quality and diversity of generated images are highly dependent on the chosen lowest resolution size. This parameter needs careful tuning for specific images at hand, as otherwise image layouts may exhibit lack of diversity or loss of global coherency (see Fig. 3). Moreover, this approach does not generalize to learning from multiple images, as in the single video case (see Table 2 and Fig. 5).

Figure 3: Limitation of the multi-stage training of single image patch-GAN methods. As the finer generation stages cannot correct the layout decision made by the coarser scale generators, without a careful tuning of the lowest resolution size the model produces images of very low diversity (first row) or lacking global coherency (last row).

We therefore introduce an alternative solution to overcome the memorization effect but still to produce high-quality images. We note that in order to produce realistic and diverse images, the generator should learn the appearance of objects and combine them in the image in a globally-coherent way. To this end, we propose a discriminator that judges the *content* distribution of a given image separately from its *layout* realism. To achieve the disentanglement, we design a two-branch discriminator architecture, with separate content and layout branches. Note that the branching of the discriminator happens after intermediate layers; this is done in order to learn relevant representations for building the branches. As seen from Fig. 2, our discriminator consists of the low-level feature

extractor $D_{low\text{-}level}$, the content branch $D_{content}$, and the layout branch $D_{layout}$. For a given image $x$, the purpose of $D_{low\text{-}level}$ is to learn low-level features and to produce an image representation $F(x) = D_{low\text{-}level}(x)$ for the branches. Next, $D_{content}$ will judge the content of $F(x)$, irrespective from its spatial layout, while on the other hand $D_{layout}$ will inspect only the spatial information extracted from $F(x)$. Inspired by the attention modules of Park et al. (2018) and Woo et al. (2018), we implement the content-layout disentanglement by squeezing channels or spatial dimensions of the intermediate features $F(x)$. Note that afterwards the branches $D_{content}$ and $D_{layout}$ receive only limited information about the image from $F(x)$, preventing them from overfitting to the whole image, and thus mitigating the negative effect of memorizing the original image.

**Content branch.** The content branch decision should be based upon the image content, i.e. the fidelity of objects composing the image, independent of their spatial location in the scene. Let the feature map $F(x)$ have dimensions $H(\text{height}) \times W(\text{width}) \times C(\text{channel})$. Note that the spatial dimensions $H \times W$ capture spatial information, while the channels $C$ encode the semantic representation. As we want the content branch to ignore the spatial location of objects, we apply global average pooling to aggregate the spatial information $H \times W$ across the channels $C$. The resulting feature map $F_{content}(x)$ has size $1 \times 1 \times C$, which is then processed by several layers for further real/fake decision making. By removing the spatial information, $D_{content}(x)$ is induced to respond to content features encoded in different channels regardless of their spatial location (see App. A.1).

**Layout branch.** The layout branch, in contrast, should assess the spatial location of objects in the scene, but not their specific appearance. Thus, the layout branch is designed to judge only the spatial information of $F(x)$, filtering out the content details. Since the layout information is encoded only in spatial dimensions $H \times W$, and not in channels $C$, we aggregate the channel information from $F(x)$ via a $(1 \times 1)$ convolution with only one output channel, which forms a feature map $F_{layout}(x)$ with size $H \times W \times 1$. This channel aggregation weakens the content representation but does not affect the spatial information. The $F_{layout}(x)$ features are further processed by several layers before a real/fake decision is made. As $D_{layout}(x)$ is designed to be sensitive only to the spatial representation of the input image, it learns to judge the realism of scene layouts (see App. A.1).

**Feature augmentation.** The proposed two-branch discriminator prevents the memorization of training samples, enabling the generation of images with content and layouts different from the original sample. To further improve the diversity of generated images, we propose to augment the content $F_{content}(x)$ and layout $F_{layout}(x)$ features of real images. For the single image setting this is done by mixing the features of two different augmentations of the original image, and for the single video setting by mixing the features of augmentations of two different video frames. For two real samples $x_1$ and $x_2$ we apply a mixing transformation $F_*(x_1) = T_{mix}(F_*(x_1), F_*(x_2))$. We use two types of mixing: 1) For the layout branch, we sample a rectangular crop of $F_{layout}(x_2)$ and paste it on to $F_{layout}(x_1)$ at the same spatial location, similarly to CutMix (Yun et al., 2019). In contrast to CutMix, our approach augments features, not input images, and mixes only features of real images. 2) For the content branch, we sample a set of channels from $F_{content}(x_2)$ and copy their values to the corresponding channels of $F_{content}(x_1)$. As the channels encode semantic features of images, we expect the resulting augmented tensor to represent objects seen in two different images. We also found it useful to remove channels from $F_{content}(x)$, thus removing some object representations. For this, we sample a set of channels and drop out their values (Srivastava et al., 2014; Zhengsu et al., 2018). With the above augmentations, $D_{content}$ and $D_{layout}$ see significantly more variance in both the content and layout representations of real images, which prevents overfitting and improves the diversity of generated samples. The feature augmentation (FA) effect is shown in Table 3.

**Adversarial loss.** To evaluate images at different scales, we design our discriminator to make a binary true/fake decision at each intermediate resolution. For each discriminator part $D_*$: $D_{low\text{-}level}$, $D_{content}$, and $D_{layout}$, the loss is computed by aggregating the contributions across all layers constituting the corresponding discriminator part:

$$\mathcal{L}_{D_*} = \frac{1}{N_*} \sum_{l=1}^{N_*} \mathcal{L}_{D_*^l}, \tag{1}$$

where $D_*^l$ is the $l$-th ResNet block of $D_*$, $N_*$ is the number of ResNet blocks used in $D_*$, and the loss $\mathcal{L}_{D_*^l}$ is the binary cross-entropy: $\mathcal{L}_{D_*^l} = -\mathbb{E}_x[\log D_*^l(x)] - \mathbb{E}_z[\log(1 - D_*^l(G(z)))]$. $D_*^l$ aims to distinguish between real $x$ and generated $G(z)$ images based on their corresponding features at block $l$, which captures either their low-level details, content, or layout at a certain resolution. The overall adversarial loss for SIV-GAN is then computed by taking the decisions from the content

branch $D_{content}$, the layout branch $D_{layout}$, and the low-level features of $D_{low\text{-}level}$:

$$\mathcal{L}_{adv}(G, D) = \mathcal{L}_{D_{content}} + \mathcal{L}_{D_{layout}} + 2\mathcal{L}_{D_{low\text{-}level}}, \qquad (2)$$

where $D$ aims to distinguish between real and generated images based on their low-level, content, and layout realism. As the two discriminator branches operate on high-level image features, contrary to only one $D_{low\text{-}level}$ operating on low-level features, we double the weighting for $\mathcal{L}_{D_{low\text{-}level}}$. This is done in order to properly balance the contributions of different feature scales and encourage the generations of images with good low-level details, plausible content, and coherent scene layouts.

## 3.2 Diversity Regularization

To improve the variability among the generated images, we propose to add a diversity regularization (DR) loss term $\mathcal{L}_{DR}$ to the SIV-GAN objective. Prior work (Yang et al., 2019; Zhao et al., 2021; Choi et al., 2020) also proposed to use diversity regularization for GAN training, but mainly to avoid mode collapse, and assuming the availability of a large training set. The regularization of Yang et al. (2019) aimed to encourage the generator to produce different outputs depending on the input latent code, in such a way that generated samples with closer latent codes should look more similar to each other, and vice versa. In contrast, our diversity regularization is tuned for synthesis in single data instance regimes. Assuming that in case of a single image or a single video we are operating in one semantic domain, the generator should produce images that are in-domain but more or less equally different from each other, and substantially different from the original training sample. Thus, in such regimes the difference of generated images should not be dependent on the distance between their latent codes, so we propose to encourage the generator to produce perceptually different image samples independent of their distance in the latent space. Mathematically, the new diversity regularization is expressed as:

$$\mathcal{L}_{DR}(G) = \mathbb{E}_{z_1, z_2}\Big[\frac{1}{L}\sum_{l=1}^{L} \|G^l(z_1) - G^l(z_2)\|)\Big], \qquad (3)$$

where $\|\cdot\|$ denotes the $L1$ norm, $G^l(z)$ indicates a feature extracted after $l$-th resolution block of the generator $G$ given input $z$, and $z_1, z_2$ are randomly sampled latent codes in the batch, i.e. $z_1, z_2 \sim N(0, 1)$. By regularizing the generator to maximize Eq. 3, we force it to produce diverse outputs for different latent codes $z$. Note that, in contrast to Yang et al. (2019); Zhao et al. (2021); Choi et al. (2020), we compute the distance between samples in the feature space of the generator. Computing the distance in the feature space results in more meaningful diversity among the generated images, as different layers of the generator capture different image semantics, inducing both high-level and low-level diversity. Computing the distance in the image space , i.e. $\mathcal{L}_{DR}(G) = \|G(z_1) - G(z_2)\|$ as in Choi et al. (2020), leads to reduced image diversity in our experiments (see Table 4).

The overall SIV-GAN objective can be written as:

$$\max_{G}\min_{D}\ \mathcal{L}_{adv}(G, D) + \lambda\mathcal{L}_{DR}(G), \qquad (4)$$

where $\lambda$ controls the strength of the diversity regularization and $\mathcal{L}_{adv}$ is the adversarial loss in Eq. 2. The proposed diversity regularization is shown to be highly-effective for SIV-GAN, while prior regularizations (Yang et al., 2019; Zhao et al., 2021; Choi et al., 2020) underperform (see Table 4).

## 3.3 Implementation and Training

The SIV-GAN generator employs ResNet blocks, similar to BigGAN (Brock et al., 2019). However, we do not use BatchNorm or self-attention. As in MSG-GAN (Karnewar & Wang, 2020), we generate images at intermediate ResNet blocks of $G$, passing them to $D_{low\text{-}level}$ to facilitate the gradient flow from the discriminator. The latent vector $z$, of length 64, is sampled from $N(0, 1)$. For diversity regularization we use the tanh activation on the features from the final convolutions of the $G$ blocks.

The SIV-GAN discriminator also uses ResNet blocks. We set $N_{low-level} = 3$, $N_{layout} = N_{content} = 4$, thus using 3 ResNet blocks before branching and 4 ResNet blocks for the content and layout branches. To enable multi-scale gradients, we incorporate images at different scales using the $\phi_{lin\_cat}$ strategy from Karnewar & Wang (2020). The proposed feature augmentation (FA) is applied with probability $0.4$ at every discriminator forward pass. We also use differentiable image augmentation (DA) (Karras et al., 2020a; Zhao et al., 2020b;a), applying translation, cropping, rotation, and horizontal flipping for real and fake images with a probability of $0.7$ at each forward pass. As in (Karras et al., 2020a), we observe no signs of leaking augmentations in the generated samples.

Table 1: Comparison with other methods in the Single Image setting on Places (Zhou et al., 2017a) and DAVIS-YFCC100M (Perazzi et al., 2016; Thomee et al., 2016) datasets.

| Method | Places | | | | | DAVIS-YFCC100M | | | | |
|---|---|---|---|---|---|---|---|---|---|---|
| | SIFID | | LPIPS↑ | Pixel ↑ | Dist. | SIFID | | LPIPS↑ | Pixel ↑ | Dist. |
| | best ↓ | mean ↓ | | Diversity | to train | best | mean | | Diversity | to train |
| SinGAN | 0.09 | 0.15 | 0.22 | 0.52 | 0.24 | 0.10 | 0.13 | 0.26 | 0.54 | 0.30 |
| ConSinGAN | 0.06 | 0.08 | 0.24 | 0.50 | 0.25 | 0.08 | 0.09 | 0.29 | 0.59 | 0.31 |
| FastGAN | 0.11 | 0.14 | 0.15 | 0.48 | 0.08 | 0.10 | 0.13 | 0.18 | 0.49 | 0.11 |
| SIV-GAN | **0.05** | **0.06** | **0.28** | **0.57** | 0.31 | **0.07** | **0.08** | **0.33** | **0.66** | 0.37 |

Figure 4: Visual comparison in the Single Image setting. Single-image GANs (Shaham et al., 2019; Hinz et al., 2021) incoherently shuffle patches (e.g. sky textures below horizon, perturbed fish contours), and few-shot FastGAN (Liu et al., 2021) reproduces the original image or its flipped version. In contrast, SIV-GAN achieves high diversity, maintaining content and layout realism.

In contrast to previous single image GANs (Shaham et al., 2019; Hinz et al., 2021), which employ a multi-stage training scheme, SIV-GAN is trained end-to-end in one stage, with the losses from Eq. 4, with $\lambda = 0.15$ for $\mathcal{L}_{DR}$ (see the ablation on $\lambda$ in App.B.1). We use spectral normalization (Miyato et al., 2018) for both $G$ and $D$, and do not use a reconstruction loss as in (Shaham et al., 2019; Hinz et al., 2021), or any other stabilization techniques. SIV-GAN is trained using the ADAM optimizer with $(\beta_1, \beta_2) = (0.5, 0.999)$, a learning rate of $0.0002$ for both $G$ and $D$, and a batch size of 5 (using different augmentations of a single image or video frames). SIV-GAN can operate at different image resolutions by simply changing the input noise shape (see App. E.2), successfully producing images of high quality and diversity at resolution $512 \times 896$ (see Fig. 1 and Fig. J for examples).

## 4 EXPERIMENTS

We evaluate SIV-GAN in the Single Image and Single Video settings, using the model configuration described in Sec. 3.3. We use 100k iterations on an image and 300k iterations on a video for training.

**Datasets.** Following Shaham et al. (2019), we evaluate the Single Image setting on 50 images from Places (Zhou et al., 2017a). In addition to their protocol, we select 15 videos from the DAVIS (Perazzi et al., 2016) and YFCC100M (Thomee et al., 2016) datasets. In the Single Video setting, we use all the frames as training images, while for the Single Image setup we use only one middle frame. The chosen videos last for 2-10 seconds and consist of 60-100 frames.

**Metrics.** To assess the quality of generated images, we measure the single FID (SIFID) (Shaham et al., 2019). Its original formulation uses InceptionV3 features before the first pooling layer at $\frac{H \times W}{4}$ resolution, which capture low-level image details. To assess high-level properties, such as scene layouts, we also propose to compute SIFID at $\frac{H \times W}{16}$ resolution. To evaluate the diversity of samples, we adopt the pixel diversity metric from (Shaham et al., 2019). To measure perceptual diversity, we also report the average LPIPS (Dosovitskiy & Brox, 2016) across pairs of generated images. To verify that the models do not simply reproduce the training set, we report average LPIPS to the nearest image in the training set, augmented in the same way as during training (Dist. to train).

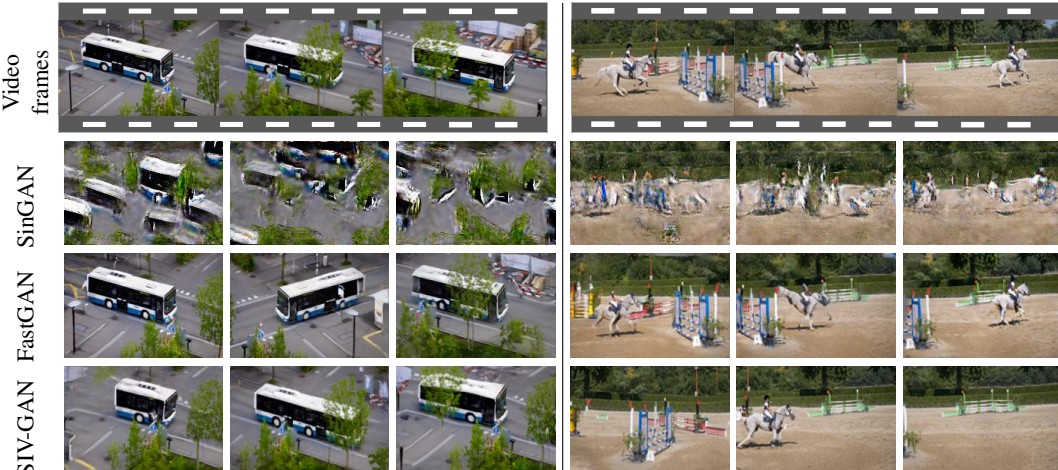

Figure 5: Visual comparison in the Single Video setting. While other models fall into reproducing the training frames or fail to correctly generate objects, SIV-GAN produces high-quality images substantially different from the original training frames.

We note that SIFID tends to penalize diversity, favouring overfitting (Robb et al., 2020). Thus, to account for this quality-diversity trade-off, a fair analysis should assess both diversity and quality.

**Comparison models.** We compare our model with single image methods, SinGAN (Shaham et al., 2019) and ConSinGAN (Hinz et al., 2021), and with a recent model for few-shot image synthesis, FastGAN (Liu et al., 2021). We use the original implementation codes provided by the authors. While training single image GANs from (Shaham et al., 2019; Hinz et al., 2021) on a single video, we applied the reconstruction loss on all frames, as we found this helpful in stabilizing the training.

## 4.1 MAIN RESULTS

Tables 1 and 2 present a quantitative comparison between the models in the Single Image and Single Video settings, while the respective visual results are shown in Fig. 4 and 5. As seen from the tables, SIV-GAN notably outperforms other models in both studied settings. Despite a potential trade-off between quality and diversity, our model achieves better performance in both, reaching lower SIFID values and higher diversity scores. Importantly, only

Table 2: Comparison in the Single Video setting on DAVIS-YFCC100M.

| Method | SIFID ↓ | | LPIPS ↑ | Dist. to train |
|---|---|---|---|---|
| | $\frac{H \times W}{4}$ | $\frac{H \times W}{16}$ | | |
| SinGAN | 2.47 | 96.35 | 0.32 | 0.51 |
| ConSinGAN | 2.74 | 74.50 | 0.34 | 0.53 |
| FastGAN | 0.79 | 9.24 | **0.43** | 0.13 |
| SIV-GAN | **0.55** | **5.14** | **0.43** | 0.34 |

SIV-GAN successfully learns from both single images and videos, generating globally-coherent images of high diversity. Next, we analyse results in these settings separately.

**Single Image.** As seen from Fig. 1 and Fig. 4, in the Single Image setting, SIV-GAN produces diverse, high-quality images. For example, our model can change the number and placement of balloons in the sky, or edit the contour and position of rocks in the desert. Note that such changes retain the appearance of objects, preserve original context, and maintain scene layout realism. In contrast, the prior single-image GAN models, SinGAN and ConSinGAN, disturb the appearance of objects (e.g. by washing away fish contours) and disrespect layouts (e.g. sky textures appear below the horizon), while having modest diversity. This is reflected in their higher SIFID and lower diversity scores in Table 1. The few-shot FastGAN model suffers from memorization, reproducing the training image or its flipped version. In Table 1 this is reflected in lowest diversity and Dist. to train (in red) metrics on both datasets. Despite having the lowest diversity, we observe that FastGAN does not reach a low SIFID due to leaking augmentations (horizontal flipping).

**Single Video.** In the Single Video setting, there is much more information to learn from, and generative models can consequently synthesize more interesting combinations of objects and scenes. Fig. 1 and Fig. 5 show the images generated by SIV-GAN in this setting. Our model generates high-quality images that are substantially different from the training frames, adding/removing objects and changing scene geometry. For example, having seen a car following a road (Fig. 1), SIV-GAN generates the scene without a car or with two cars. In Fig. 5 our model varies the length of a bus and placement of trees, or removes a horse from the scene and changes the jumping obstacle config-

Table 3: Ablation study in the Single Image and Video settings on DAVIS-YFCC100M. Indicators of collapsed diversity (low LPIPS, Pixel Diversity) or poor quality (high SIFID) are marked in red.

| | Single Image | | | | | Single Video | | | |
|---|---|---|---|---|---|---|---|---|---|
| Method | SIFID ↓ | | LPIPS ↑ | Pixel ↑ | Dist. | SIFID ↓ | | LPIPS ↑ | Dist. |
| | $\frac{H \times W}{4}$ | $\frac{H \times W}{16}$ | | diversity | to train | $\frac{H \times W}{4}$ | $\frac{H \times W}{16}$ | | to train |
| Full model | 0.08 | 16.30 | 0.33 | 0.66 | 0.37 | 0.55 | 5.14 | **0.43** | 0.34 |
| No Layout br. | 0.14 | 20.29 | **0.35** | **0.67** | 0.40 | 0.71 | 11.70 | 0.42 | 0.38 |
| No Content br. | 0.08 | 23.25 | 0.34 | 0.64 | 0.36 | 0.73 | 10.43 | 0.41 | 0.33 |
| No branches | **0.03** | **7.73** | 0.13 | 0.43 | 0.12 | 0.42 | **3.73** | 0.37 | 0.18 |
| No DR | 0.05 | 11.99 | 0.04 | 0.33 | 0.06 | **0.40** | 9.81 | 0.30 | 0.32 |
| No FA | 0.08 | 14.81 | 0.27 | 0.58 | 0.33 | 0.51 | 4.85 | 0.41 | 0.32 |
| No $\mathcal{L}_{D_{low-level}}$ | 0.08 | 15.92 | 0.27 | 0.56 | 0.29 | 0.58 | 5.32 | 0.40 | 0.31 |

uration. In contrast, SinGAN, tuned to learn from a single image, does not generalize to the Single Video setting, distorting objects and producing unrealistic scene layouts (low diversity and an extremely high SIFID in Table 2). FastGAN, on the other hand, generates images with reasonable fidelity, but fails to produce samples with non-trivial layout changes, having a very low Dist. to train score (0.13 in Table 2). Additional qualitative results can be found in App. D.1.

## 4.2 ABLATIONS

In Table 3 we demonstrate the importance of the proposed modifications. In each line we remove only one model component, starting from our full model. Firstly, we ablate our discriminator architecture, testing it without any branches (No branches), corresponding to a standard GAN discriminator, and without the layout branch or the content branch. The model without branches is trained together with our proposed DR and FA, as well as differentiable augmentations (DA) as in (Karras et al., 2020a; Zhao et al., 2020a). However, as seen from Table 3, it

Table 4: Comparison of diversity regularization techniques in the Single Image setting on DAVIS-YFCC100M.

| Regularization | SIFID ↓ | LPIPS ↑ | Pixel ↑ Diversity |
|---|---|---|---|
| None | **0.05** | 0.04 | 0.33 |
| zCR | 0.05 | 0.06 | 0.37 |
| DS | 0.06 | 0.14 | 0.45 |
| DR (im. space) | 0.07 | 0.21 | 0.52 |
| DR | 0.08 | **0.33** | **0.66** |

memorizes the training images and reproduces them with poor diversity. Using only one of the branches shows good diversity, but the model fails to generate globally-coherent images, having a high $\frac{H \times W}{16}$ SIFID. The qualitative results for these ablation models, as well as additional analysis of branches learning content and layout, are presented in App. A.1. Next, we observe the effect of the proposed diversity regularization (DR), feature augmentation (FA), and low-level loss $\mathcal{L}_{D_{low-level}}$. Without DR, the model does not achieve multi-modality, scoring low on diversity metrics. The absence of FA notably decreases diversity, which is reflected in the diversity scores dropping by 0.02-0.08 points. Removing $\mathcal{L}_{D_{low-level}}$ also reduces diversity; we discuss this effect in App. A.4. Overall, we observe that the success of our model in both settings is enabled by both the two-branch discriminator and the diversity regularization. In App. C we also show the benefit of using SIV-GAN discriminator for few-shot image synthesis, outperforming the FastGAN model.

In Table 4 we compare our DR to the latent consistency regularization (zCR) Zhao et al. (2021) and diversity-sensitive loss (DS) Yang et al. (2019). Our DR noticeably improves over zCR and DS on all diversity metrics. We find it beneficial to use DR in the feature space, which leads to more variation in the generated samples. Interestingly, Table 4 illustrates a quality-diversity trade-off, where improvements in diversity lead to deterioration in SIFID. See more ablations in App. B.

## 5 CONCLUSION

We propose SIV-GAN, a new unconditional generative model which successfully learns from a single image or a single video. In such extremely low-data regimes, our model prevents memorization and generates diverse images that are significantly different from the training set. Inherently, the synthesis of our model is constrained by the appearance of objects present in the original sample. Nevertheless, SIV-GAN can synthesize novel scene compositions by blending objects in different combinations, changing their shape or position, while preserving the original context and plausibility of the scene. Such compositionality is enabled by the model's ability to distinguish objects and backgrounds, astonishingly, while learning from a single image or video. We believe SIV-GAN provides a useful tool for data augmentation in domains where data collection remains challenging.

## ETHICS STATEMENT

All the authors of this work adhered to the ICLR Code of Ethics at the time of preparing the submission. We have made our best effort to uphold high standards of scientific excellence, avoid harm, discrimination, and dishonesty. All the experiments presented in the paper are conducted on publicly available datasets with a granted right for research-purposed usage. Our organization is carbon neutral so that all its activities including research activities do no longer leave a carbon footprint. This also includes our GPU clusters on which the experiments have been performed.

The method presented in this work allows high-quality image synthesis in extremely low data regimes. Our model allows generating novel scene compositions requiring only a single image or video for training. This ability can be potentially utilized in different applications, ranging from AI-based content editing to data augmentation of limited domains. For the content creation applications, similarly to other data-driven editing tools and graphical engines, a potential concern of misuse is the arising possibility for data manipulation. Photorealistic fake images synthesized by GANs may be presented as real photography in the media coverage, which may lead to negative societal outcomes. For the data augmentation, our model trained on rare samples can be used to generate more examples of underrepresented data, thus upsampling its proportion in the original data collection. This step will help to weaken the bias of datasets, providing more fairness in data coverage.

## REPRODUCIBILITY STATEMENT

The authors have made their best effort to ensure the reproducibility of the methods presented in this work. A complete description of our algorithmic proposals is presented in Sec. 3, and our experimental and evaluation settings are discussed in Sec. 4. An extended description of all architectural solutions, training details, including a detailed scheme of our networks is included in App. E. To facilitate the usage of our models and the reproduction of our results, we will release publicly our training code and trained model weights, providing a complete list of images and videos used for experiments.

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

APPENDIX

This supplementary material is structured as follows:

# A ADDITIONAL ANALYSIS OF THE SIV-GAN DISCRIMINATOR

## A.1 EFFECT OF THE TWO BRANCHES IN THE DISCRIMINATOR

In Fig. A and B we showcase the visual effect of using content and layout branches in the SIV-GAN discriminator. Following the ablation Table 3 in the main paper, we separately show qualitative results for our full discriminator model with content and layout branches, the discriminator without the layout branch, without the content branch, and for a model without both branches, corresponding to a standard GAN discriminator. The visual results illustrate the concepts of "layout" and "content" learnt by the respective branches. Applying the content branch, without the layout branch, leads to the generation of different objects in various combinations, but the model often fails to reproduce correct positioning of objects or globally-coherent layouts. For example, there might be a horizon discontinuity, or air balloons may follow unrealistically structured positions in a grid. In contrast, a model, trained with the layout branch, but without the content branch, generates images with more realistic layouts, but does not preserve the content distribution of the training image, removing objects, distorting their appearance or perturbing their shapes. Behaviour of such models corresponds to high diversity scores and poor low-level SIFID in Table 3. Employing none of the branches, which corresponds to using a standard GAN discriminator, leads to memorization of the training image, so the model just reproduces the training data (low distance to training set in Table 3). Finally, our full model with a two-branch discriminator generates plausible diverse images, varying both the image content and the global layout of the scene (0.33 LPIPS and SIFID at scale $\frac{H \times W}{16}$ of 16.3 in the Single Image setting, see Table 3).

To illustrate further our intuition on the discriminator's branches learning content and layout, we analyse the feature distances between real images in the content and layout embeddings of the trained SIV-GAN discriminator. For this, we take the discriminators trained on single "bus" and "parkour" videos from the DAVIS dataset (Perazzi et al., 2016). The video sequences have notably different variability in layouts: while there is a significant camera direction movement for the "parkour"

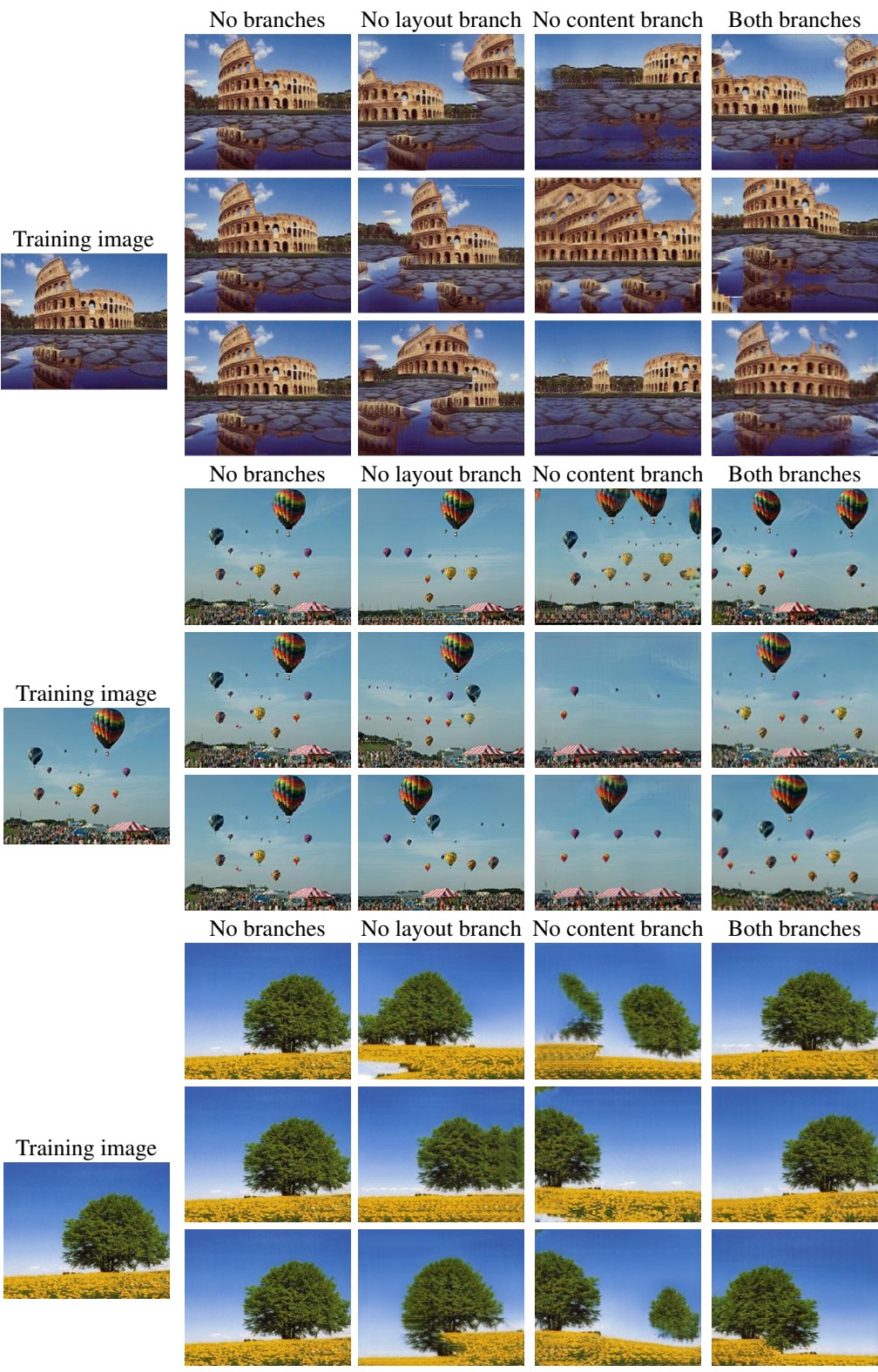

Figure A: Visual effects of using content and layout branches in the discriminator in the Single Image setting. The model with a standard GAN discriminator (No branches) memorizes the training image. Model without the layout branch fails to produce images with realistic layouts or positioning of objects. Absence of the content branch does not preserve well object appearances. Finally, the model with both branches generates diverse images with realistic content and layouts. See Table 3 for qualitative comparison and App. A.1 for the discussion.

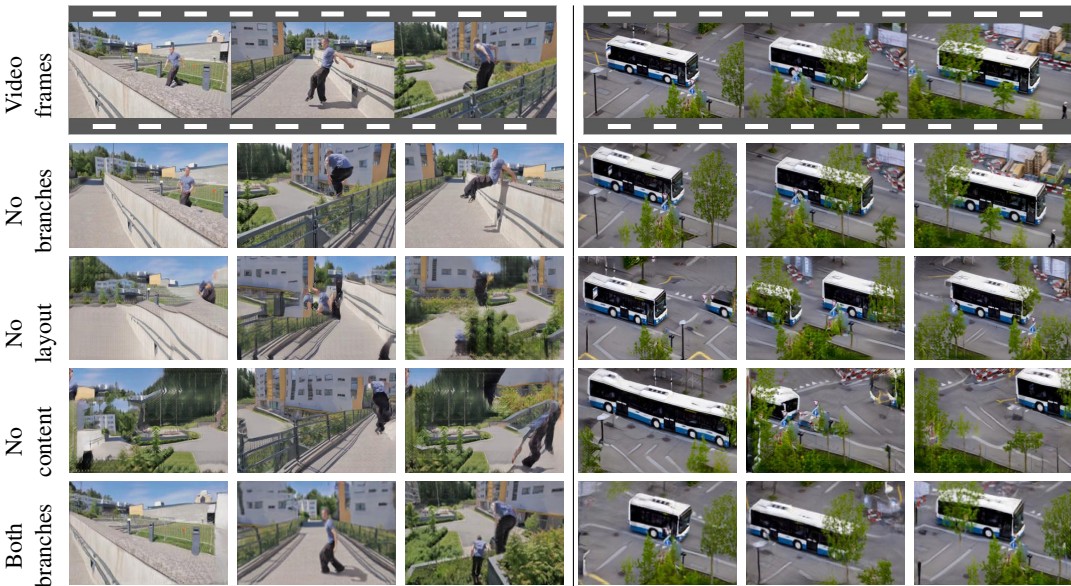

Figure B: Visual effects of using content and layout branches in the discriminator in the Single Video setting. The model with a standard GAN discriminator (No branches) memorizes the training video frames. The model without the layout branch is prone to cutting objects' contours, generating globally incoherent layouts. The absence of the content branch leads to failures in appearances of objects, such as trees. Finally, the model with both branches generates diverse images maintaining realistic content and layouts. See Table 3 for qualitative comparison and App. A.1 for the discussion.

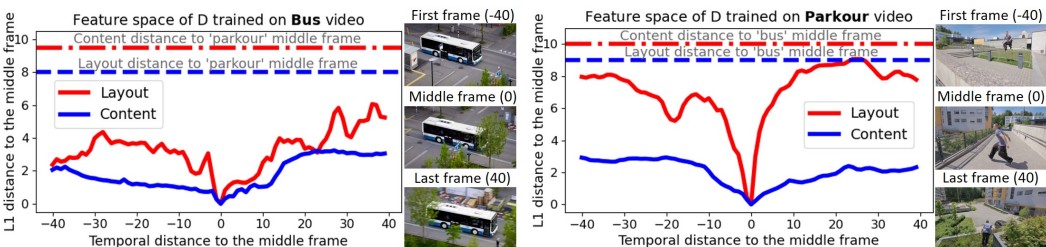

Figure C: Feature distances from video frames to middle frame. As nearby frames have very similar content and layouts, the lowest distances are between adjacent frames. The layout embedding distances (solid red) between bus frames are lower than for the parkour video, being in line with the visual layout variance of these sequences. As frames in the short videos depict similar content (e.g. same objects), the content embedding distances between frames of the same sequence (solid blue) are significantly lower than distance between middle frames of different videos (dashed blue).

video, the "bus" video captures the vehicle always from the same angle. This leads to a much smaller perceptual differences in layouts for the bus video compared to the parkour video.

In Fig. C we show the feature distances in the embeddings of SIV-GAN content and layout branches, measured between the middle frame and other frames of the same video sequence. We observe that the illustrated distances correlate well with our intuition of $D$ branches learning content and layout. Firstly, as nearby frames have very similar content and layouts, the lowest distances both in the content and layout embeddings are between adjacent frames, while increasing the temporal distance from the middle frames leads to higher content and layout feature distances. Secondly, the embedding differences are consistent with our perceptual judgement: the layout embedding distances (solid red) between "bus" frames are notably lower than for the "parkour" video. Finally, as video frames inside the short videos depict similar content (e.g. same objects), the content embedding distances between frames of the same sequence (solid blue) are significantly lower than to middle frames of two different videos (dashed blue).

Training frame $N_{D_{low\text{-}level}}=1$ $N_{D_{low\text{-}level}}=2$ $N_{D_{low\text{-}level}}=3$ $N_{D_{low\text{-}level}}=4$ $N_{D_{low\text{-}level}}=5,6$

Figure D: Effect of the number of discriminator blocks used before branching. Using too few blocks (1-2) leads to reduced synthesis quality, as $D_{low\text{-}level}$ is unable to extract the features necessary to build the content and layout representations. Increasing the number of blocks (3-4) results in improved quality while maintaining good diversity. Using too many blocks (5-6) leads to the memorization effect due to the occurring overfitting, when the model reproduces the training image with no diversity.

Table A: Ablation on the number of blocks $N_{D_{low\text{-}level}}$ used before content-layout branching in the discriminator on the DAVIS-YFCC100M dataset.

| $N_{D_{low\text{-}level}}$ | Single Image | | Single Video | |
|:---:|:---:|:---:|:---:|:---:|
| | SIFID ↓ | LPIPS ↑ | SIFID ↓ | LPIPS ↑ |
| 1 | 0.59 | **0.42** | 2.75 | **0.46** |
| 2 | 0.13 | 0.40 | 1.12 | 0.45 |
| 3 | 0.08 | 0.33 | 0.55 | 0.43 |
| 4 | 0.06 | 0.24 | 0.36 | 0.38 |
| 5 | **0.03** | 0.15 | 0.40 | 0.37 |
| 6 | **0.03** | 0.13 | **0.35** | 0.36 |

## A.2 ABLATION ON THE NUMBER OF BLOCKS USED BEFORE BRANCHING

The proposed SIV-GAN discriminator has two branches, preceded by a low-level feature extractor. As discussed in Sec. 3.3, the discriminator consists of 7 ResNet blocks, using 3 ResNet blocks before the branching for $D_{low\text{-}level}$, and 4 ResNet blocks for the both branches $D_{content}$ and $D_{layout}$. In Table A and Fig. D we analyse the effect of applying branching at an earlier or a later discrimination stage, keeping the overall depth of the networks equal to 7 ResNet block. The results indicate that the branching should be applied neither too early nor too late. Using too few ResNet blocks (1-2) before the branching leads to a reduced capacity of the low-level feature extractor $D_{low\text{-}level}$, so this network becomes unable to provide the branches with descriptive content and layout representations of an image. As seen from Fig. D, such model learns the color distribution of an image, but cannot produce a globally-coherent scene and generate textures of good quality. In Table A this effect is indicated by a very high SIFID. On the other hand, using too many blocks before the branching (5-6) leads to increased capacity of the feature extractor $D_{low\text{-}level}$, so the network can remember the whole image. In this case the model suffers from a memorization effect, reproducing the training image (see Fig. D), and scoring low at diversity metrics (low LPIPS in Table A). Finally, we found that using 3 ResNet blocks before the branching leads to an optimal quality-diversity trade-off in both the Single Image and Single Video settings, allowing the generation of high diversity, while preserving the context of original samples.

## A.3 EFFECT OF THE NUMBER OF CHANNELS IN THE LAYOUT BRANCH

In this section we study the effect of varying the number of channels in the layout branch feature representation. For all experiments in the main paper, the layout branch representation is obtained from an intermediate representation by a convolutional layer with one output channel. Table B shows the effect of using the layout representation with 3, 5, and 10 channels. An increased number of channels in the layout branch results in a more powerful network that can easier memorize the layout representations extracted from real images. Thus, increasing the number of channels results in reduced diversity of samples, as indicated by the smaller LPIPS scores in Table B.

## A.4 EFFECT OF THE LOW-LEVEL LOSS

In this section we provide an ablation on the low-level discriminator loss $\mathcal{L}_{D_{low\text{-}level}}$. According to Eq. 1 and 2, the loss of the SIV-GAN discriminator is computed after each ResNet block before branching, which means that the attention of the discriminator is shared across different layers, corresponding to image realism at different low-level scales. Without this loss, the discriminator

Table B: Ablation on the number of channels used for the layout branch feature representation on the DAVIS-YFCC100M dataset.

| $F_{layout}$ channels | Single Image SIFID ↓ | LPIPS ↑ | Single Video SIFID ↓ | LPIPS ↑ |
|---|---|---|---|---|
| 1 | 0.08 | **0.33** | 0.55 | **0.43** |
| 3 | 0.07 | 0.28 | 0.54 | 0.41 |
| 5 | **0.06** | 0.25 | 0.50 | 0.40 |
| 10 | **0.06** | 0.24 | **0.49** | 0.40 |

Table C: Ablation on the low-level loss $\mathcal{L}_{D_{low-level}}$ on DAVIS-YFCC100M dataset.

| Low-level loss | Single Image SIFID ↓ | LPIPS ↑ | Single Video SIFID ↓ | LPIPS ↑ |
|---|---|---|---|---|
| ✗ | **0.08** | 0.27 | 0.58 | 0.40 |
| ✓ | **0.08** | **0.33** | **0.55** | **0.43** |

judges images only after branching, paying attention mostly to the high-level image realism. The low-level loss changes the discriminator's task and shifts its attention towards earlier layers, thus increasing its attention to low-level details, e.g., such as textures. As seen from Table C, this property helps to reduce overfitting and allows synthesis of higher diversity, both in the Single Image and Single Video settings.

# B   ADDITIONAL ANALYSIS OF DIVERSITY REGULARIZATION

## B.1   ABLATION ON THE STRENGTH OF THE PROPOSED DIVERSITY REGULARIZATION

The proposed diversity regularization (DR) is an essential component for SIV-GAN to achieve high diversity among generated samples. For all our experiments in the main paper we used DR with $\lambda = 0.15$. In Table D, we show the effect of setting different $\lambda$ for DR in the Single Image setting, changing the strength of the diversity regularization. We note a general quality-diversity trade-off which is present for the Single Image setting. While diversity metrics favour diverse multi-modal outputs, the image quality metrics, such as SIFID, are computed based on similarity to the training frame, so they penalize variations from the original image (Robb et al., 2020). Table D illustrates that setting the multiplier too high ($\lambda = 0.50$) leads to good diversity, but harms image quality, while setting small values ($\lambda = [0.00, 0.05]$) is beneficial for quality, but deteriorates diversity. We observed that using $\lambda = 0.15$ leads to a good trade-off, resulting in a high diversity among generated samples, while not corrupting the quality of textures and the global layout coherency, so we picked this value for the final version of the model.

## B.2   EFFECT OF THE PROPOSED DIVERSITY REGULARIZATION ON OTHER GAN MODELS

To examine the effect of the proposed DR on other models, we trained SinGAN, ConSinGAN and FastGAN with the DR added to the GAN objective. Following our design, this loss was computed in the feature space, as in Eq. 3. When training the multi-stage single-image GANs from Shaham et al. (2019); Hinz et al. (2021), we applied the DR during training of each stage. To select $\lambda$, we tested the models with the values from Table D $(0.05, 0.15, 0.50)$, and chose the highest coefficient that improved diversity but did not lead to a very high SIFID, indicating reduced low-level quality of generated images.

Table D: Effect of the diversity regularization (DR) strength in the Single Image setting on DAVIS-YFCC100M.

| DR $\lambda$ | SIFID ↓ | LPIPS ↑ | MS-SSIM ↓ | Pixel ↑ Diversity | Dist to train |
|---|---|---|---|---|---|
| 0.00 | **0.05** | 0.04 | 0.95 | 0.33 | 0.06 |
| 0.05 | 0.07 | 0.19 | 0.77 | 0.50 | 0.23 |
| 0.15 | 0.08 | 0.33 | 0.63 | 0.66 | 0.37 |
| 0.50 | 0.13 | **0.39** | **0.55** | **0.69** | **0.46** |

Table E: Effect of DR applied to different GAN models in the Single Image setting on DAVIS-YFCC100M.

| Method | DR $\lambda$ | SIFID $\downarrow$ | LPIPS $\uparrow$ | MS-SSIM $\downarrow$ | Pixel $\uparrow$ Diversity | Dist. to train |
|---|---|---|---|---|---|---|
| SinGAN | - | **0.13** | **0.26** | 0.69 | 0.54 | 0.24 |
| | 0.15 | 0.15 | **0.26** | **0.68** | **0.55** | **0.26** |
| ConSinGAN | - | **0.09** | 0.29 | 0.65 | 0.59 | 0.25 |
| | 0.05 | 0.11 | **0.30** | **0.63** | **0.60** | **0.28** |
| FastGAN | - | **0.13** | 0.18 | 0.77 | 0.49 | 0.08 |
| | 0.15 | 0.14 | **0.21** | **0.74** | **0.52** | **0.13** |
| SIV-GAN | - | **0.05** | 0.04 | 0.95 | 0.33 | 0.06 |
| | 0.15 | 0.08 | **0.33** | **0.63** | **0.66** | **0.31** |

The performance of the models with DR, as well as our selected $\lambda$ values are included to Table E. As seen from the table, DR plays the biggest role in combination with SIV-GAN, while applying DR to other methods leads only to minor improvement in diversity ($0.01 - 0.03$ MS-SSIM). For the single image GANs (Shaham et al., 2019; Hinz et al., 2021) this is explained by their multi-stage training schemes: their discriminators already overfit to all possible patches at a given scale, so the DR does not help to learn more diverse combinations. Single branch discriminator of the few-shot model (Liu et al., 2021) overfits easily when trained with very little data, and DR alone cannot correct this. We conclude that our two-branch discriminator is crucial to leverage the benefit of the proposed diversity regularization.

## C RESULTS ON FEW-SHOT IMAGE SYNTHESIS

### C.1 STANDARD FEW-SHOT IMAGE SYNTHESIS BENCHMARKS

In addition to the Single Image and Single Video settings, we conduct experiments also for few-shot image synthesis. Under this scenario, the task is to train a generative model on a small dataset, typically containing up to several hundreds of images. We experiment on the commonly used 100-shot datasets from (Zhao et al., 2020a) (Obama, Grumpy Cat, Panda) and on 160- and 389-shot datasets from (Si & Zhu, 2011) (Face-Cat, Face-Dog). These datasets have same image resolution of 256x256 and share a common object-centric structure, depicting faces of humans or animals which are centred in all images.

Many methods address few-shot image synthesis in the paradigm of few-shot adaptation, adopting GAN models pre-trained on larger datasets (Wang et al., 2018; Mo et al., 2020; Noguchi & Harada, 2019; Wang et al., 2020)). Another line of work proposes to stabilize few-shot GAN training from scratch, by utilizing differentiable image augmentation (DA) and LeCam consistency regularization (CR) (Tseng et al., 2021), or by using a skip-layer excitation module (SLE) for faster learning of the generator with a self-supervised discriminator (Liu et al., 2021)). The two most recent studies, LeCam CR (Tseng et al., 2021) and FastGAN (Liu et al., 2021), demonstrated that it is possible to outperform few-shot adaptation methods by using GAN models without pre-training, achieving significantly better performance on the standard few-shot image synthesis benchmarks (see Table F). We select FastGAN (Liu et al., 2021) as our main comparison model as it is the state-of-the-art model with the official implementation code available in open source [1].

To demonstrate the benefit of the SIV-GAN model design on the few-shot image synthesis task we apply the following modifications. Since in the few-shot setting the training set contains multiple data instances, we need to increase the capacity of our generator by increasing its depth by one ResNet block and increasing the channel multiplier (overall increasing to $\sim$30M parameters from the original 5M). In order to enable a fast learning of the larger generator we adopt the SLE module of Liu et al. (2021). We make two changes to the configuration of our discriminator. Firstly, to closer match the learning capacity of a more heavy-weight generator, we increase the width of our discriminator by doubling the channel multiplier in all layers. This step results in an increase of trainable parameters count from 1.6M to 5M. Secondly, we move the branching point to a later stage, setting $N_{\text{low-level}} = 6$. This step is necessary to follow the structure of object-centric few-

---

[1] https://github.com/odegeasslbc/FastGAN-pytorch

Table F: Results on the few-shot image synthesis task. The FID is computed between 5000 generated images and the whole training set. **Bold** and underlined indicate first and second best scores. Our SIV-GAN+ achieves better FID scores compared to FastGAN on all few-shot datasets, also outperforming the model from LeCam (Tseng et al., 2021) on four out of five datasets.

| FID ↓ | Dataset: | Obama | Grumpy Cat | Panda | Face Cat | Face Dog |
|---|---|---|---|---|---|---|
| | Number of images: | 100 | 100 | 100 | 160 | 389 |
| Scale/Shift (Noguchi & Harada, 2019) ↻ | | 50.72 | 34.20 | 21.38 | 54.83 | 83.04 |
| MineGAN (Wang et al., 2020) ↻ | | 50.63 | 34.54 | 14.84 | 54.45 | 93.08 |
| TransferGAN (Wang et al., 2018) ↻ | | 48.73 | 34.06 | 23.20 | 52.61 | 82.38 |
| FreezeD (Mo et al., 2020) ↻ | | 41.87 | 31.22 | 17.95 | 47.70 | 70.46 |
| LeCam (Tseng et al., 2021) | | **33.16** | 24.93 | 10.16 | 34.18 | 54.88 |
| FastGAN (Liu et al., 2021) | | 38.59 | 27.08 | 9.63 | 33.50 | 53.39 |
| SIV-GAN+ | | 35.10 | **23.79** | **9.22** | **31.24** | **50.05** |

↻ - models which use pre-training

Table G: Results in the extremely low few-shot data setting, where the models are trained only on subsets of standard few-shot datasets. Our model outperforms the baseline in all data regimes both in quality and diversity, and does not suffer from training instabilities even in extreme cases, such as using only 25% of the training set. Collapsed runs with a high FID for Fast-GAN are shown in red.

| Method | Panda (100) | | | | | | Face Dog (389) | | | | | |
|---|---|---|---|---|---|---|---|---|---|---|---|---|
| | 100% | | 50% | | 25% | | 100% | | 50% | | 25% | |
| | FID | LPIPS | FID | LPIPS | FID | LPIPS | FID | LPIPS | FID | LPIPS | FID | LPIPS |
| FastGAN | 9.63 | 0.48 | 13.97 | 0.49 | 16.08 | 0.47 | 53.39 | 0.62 | 77.57 | 0.61 | 96.86 | 0.61 |
| SIV-GAN+ | **9.22** | **0.51** | **11.57** | **0.51** | **9.58** | **0.50** | **50.05** | **0.63** | **64.21** | **0.62** | **62.61** | **0.62** |

shot datasets. As objects in such datasets cover whole images, we observed that the layout decision should be taken at a more global scale (4x4 instead of the previous 32x32 spatial resolution for the layout representation), while the content representation should have a large enough receptive field to see the complete object. Finally, in line with (Liu et al., 2021), we keep the self-supervision loss for discriminator, as we found it to stabilize the training in the few-shot data regime. Further we refer to our extended model as SIV-GAN+.

We train FastGAN and SIV-GAN+ for 100k iterations. We save the checkpoints every 10k iterations during training and report the best FID across all the checkpoints. The FID is computed between 5000 generated images and the whole reference dataset. The results are shown in Table F. We generally manage to reproduce the FID performance of FastGAN reported in (Liu et al., 2021), obtaining slightly better results for all datasets. Table F shows that our SIV-GAN+ model outperforms Fast-GAN, achieving an average improvement of 8% in FID across all datasets. Moreover, SIV-GAN+ also shows better results than the recent GAN model of Tseng et al. (2021), yielding the best FID on four out of five commonly used few-shot image synthesis datasets. We show the visual comparison of our model with FastGAN in Fig. E (two upper rows). Our model achieves a notably higher-quality synthesis on the Dog Face dataset, where FastGAN suffers from instabilities, having problems in generating good textures and correct shapes of dog faces.

## C.2  EXTREMELY LOW FEW-SHOT SETTING

Training GANs in low data regimes is challenging, because a GAN discriminator is prone to overfitting and memorization effects, which can result in the divergence of the training progress. Existing state-of-the-art few-shot models (Tseng et al., 2021; Liu et al., 2021) were demonstrated to succeed in this task on datasets containing at least 100 images (see App. C.1). In this section, we investigate more extreme scenarios with datasets consisting of less than 100 images. We construct "extremely low" few-shot learning regimes by selecting subsets of Obama and Face-Dog datasets containing only 50% and 25% of original training images.

The quantitative results for the extremely low few-shot settings are shown in Table G, while the visual results on 25%-subsets is shown in Fig. E (two bottom rows). As seen from the figure, the

FastGAN, trained on **full dataset** (389 images)   SIV-GAN+, trained on **full dataset** (389 images)

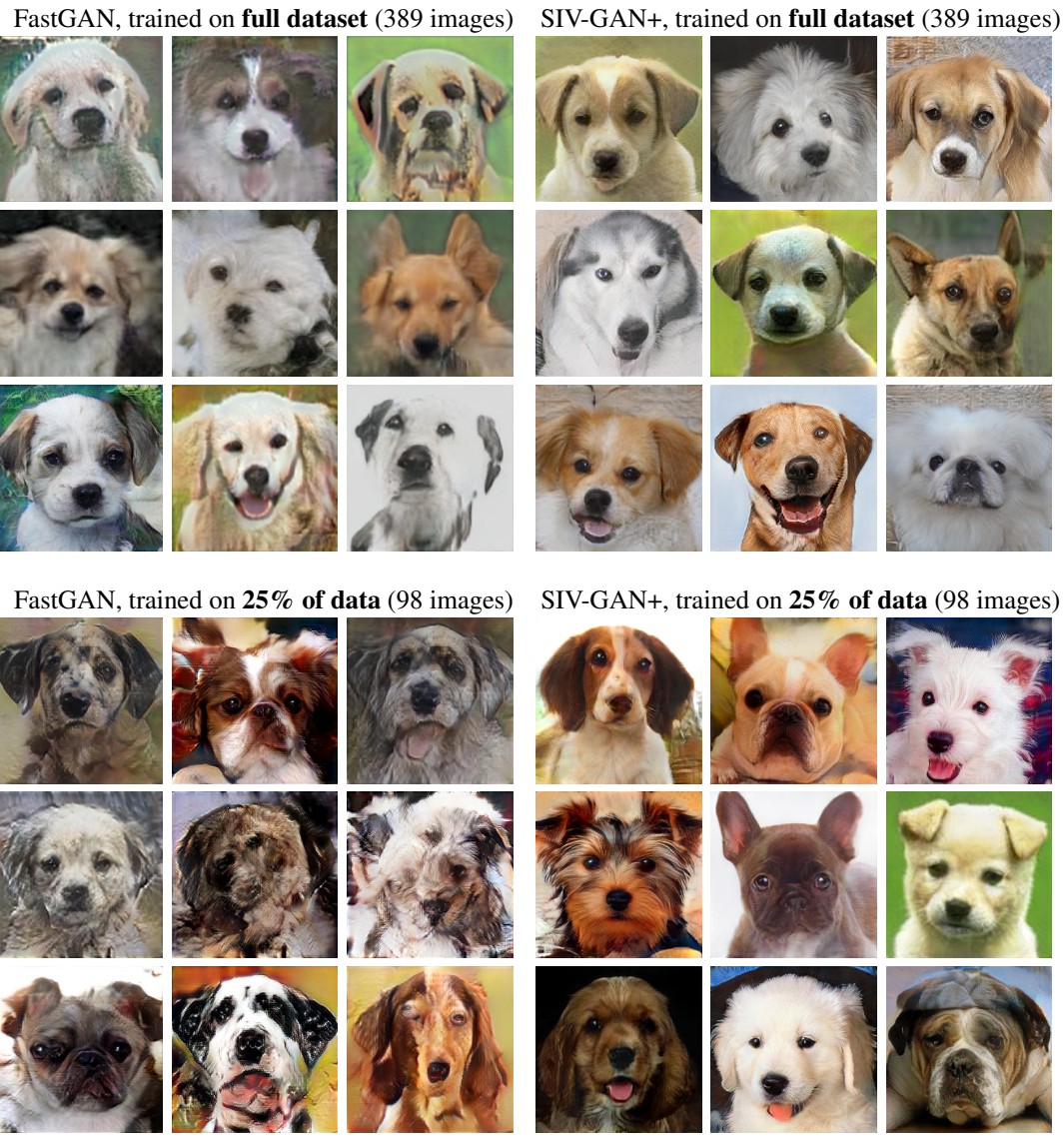

FastGAN, trained on **25% of data** (98 images)   SIV-GAN+, trained on **25% of data** (98 images)

Figure E: Visual results in the few-shot setting on the Face Dog dataset. The two upper blocks show generated images for the models trained on the full dataset (389 images), while the two bottom blocks show the results from training only using 25% of data (98 images). SIV-GAN+ demonstrates not only superior performance on the standard few-shot setting, but also successfully deals with an extreme few-shot scenario (25% of data), where FastGAN has a significant drop in performance.

FastGAN model experiences a drop in performance in the more extreme data setting, when using less than 100 images for training, producing unrealistic textures or incoherent shapes of dog faces. In Table G this effect is reflected in high FID scores (highlighted in red). In contrast, SIV-GAN+ shows good image quality uniformly across settings, reaching good FID even on training sets containing very few images. Importantly, the gain in FID is not achieved at the cost of diversity: our model has a higher LPIPS diversity score in all data regimes compared to the baseline. Moreover, Table G shows similar diversity scores for SIV-GAN+ in various settings, indicating that our model can be successfully scaled to various extremely low-data regimes maintaining similar levels of synthesis diversity. Overall, the results demonstrate that our proposed two-branch discriminator helps to mitigate discriminator overfitting in few-shot regimes, stabilizing the training, and thus helping to maintain good quality and diversity even when the model is trained on datasets with very few images.

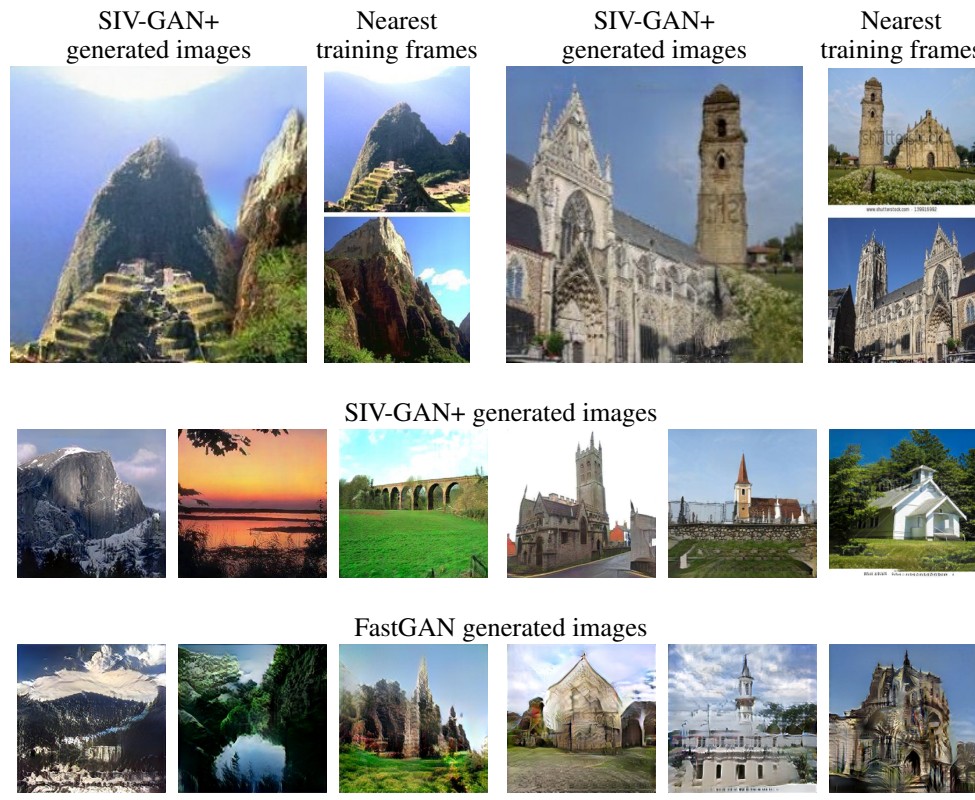

Figure F: Visual examples in the few-shot setting on the ADE-Outdoors and LSUN Church datasets. Despite being trained only on 100 images, SIV-GAN+ does not suffer from memorization issues and achieves an astonishing capability to combine objects from different training images, preserving the global coherency of scenes. In contrast, FastGAN does not achieve high-quality synthesis, struggling to generate coherent images with fine textures.

Table H: Results on few-shot datasets with a non-object-centric structure. Our model outperforms FastGAN both in quality and diversity on both datasets.

| Method | ADE-Outdoor (100) | | | | | | LSUN Church (100) | | | | | |
|---|---|---|---|---|---|---|---|---|---|---|---|---|
| | 100% | | 50% | | 25% | | 100% | | 50% | | 25% | |
| | FID | LPIPS | FID | LPIPS | FID | LPIPS | FID | LPIPS | FID | LPIPS | FID | LPIPS |
| FastGAN | 113.2 | 0.62 | 125.1 | 0.62 | 98.28 | 0.61 | 90.65 | 0.58 | 121.4 | 0.58 | 114.1 | 0.56 |
| SIV-GAN+ | **88.46** | **0.64** | **82.89** | **0.63** | **52.96** | **0.63** | **85.01** | **0.60** | **103.8** | **0.60** | **90.10** | **0.59** |

## C.3 NON-OBJECT-CENTRIC FEW-SHOT IMAGE SYNTHESIS

As mentioned in App. C.1, standard few-shot image synthesis benchmarks include only object-centric datasets, such as centred faces of people or animals. Such datasets have very limited variability in layouts, having very similar locations of face parts in all images. In this section, we explore few-shot image synthesis on datasets with more complex structures. For this, we construct subsets of the ADE-Outdoors (Zhou et al., 2017b) and LSUN-Church (Yu et al., 2015) datasets, consisting of 100 images with resolution 256x256. Such datasets form a more difficult task, because the models have to learn how to combine objects with different semantics from different images preserving the global scene layout.

The quantitative comparison between SIV-GAN+ and FastGAN is shown in Table H. Our model outperforms FastGAN in different data regimes (100, 50, 25 images) in both quality and diversity. Notably, the structure of the ADE-Outdoors and LSUN-Church datasets provides potential to generate novel compositions of objects which were not seen in the training data. We show examples of

interesting scene compositions produced by our model in Fig. F). Being trained only on 100 images of outdoor landscapes or churches, SIV-GAN+ does not memorize the training examples, generating novel compositions of mountains or exchanging towers of different churches. In the meantime, FastGAN suffers from instabilities in this setting, struggling to generate images with fine textures.

# D  ADDITIONAL QUALITATIVE AND QUANTITATIVE RESULTS

## D.1  QUALITATIVE RESULTS IN THE SINGLE IMAGE AND THE SINGLE VIDEO SETTINGS

In Fig. G, Fig. H and Fig. I we present additional qualitative results. In Fig. G and Fig. H we compare SIV-GAN to other models in the Single Image setting on the Places dataset (Zhou et al., 2017a). We note that the previous single-image methods, SinGAN and ConSinGAN, tend to shuffle image patches in a globally-incoherent way, as object textures may leak into surfaces of other semantic regions. Moreover, such methods do not preserve the appearance of objects, for example, by washing away sculptures in the garden or a man in the forest. For images with complex scenes these methods do not provide noticeable diversity, as in the example with a children playground. On the other hand, FastGAN (Liu et al., 2021) (see Fig. H) suffers from memorization, falling into reproducing a training image or its flipped version. In contrast, our SIV-GAN preserves objects, maintains global layout coherency and produces images with significant variability.

Fig. I shows the images generated by SIV-GAN in the Single Video setting. Compared to the Single Image setting, in this scenario there is much more data to learn from, and generative models can learn more interesting combinations of objects in the scenes. As seen from the figure, our SIV-GAN manages to preserve the context of the training frames, at the same time adding non-trivial semantic changes to original scenes. For example, for the shown videos, the generated frames can have a different number of balloons in sky, and combinations of planes, boats or buildings that were not seen during training.

## D.2  HIGH RESOLUTION IMAGE SYNTHESIS RESULTS

The experiments described in Sec. 4 were conducted at the image resolution of 192x320. In this section, we demonstrate the ability of our model to generate images at a higher resolution 512x896 on the DAVIS-YFCC100M dataset. For this, we add one ResNet block to the generator and discriminator, and change the input noise shape from 3x5 to 4x7 (see App. E for more architectural details). After this change, the model produces images at a much higher

Table I: Comparison of synthesis quality and diversity at different image resolutions on DAVIS-YFCC100M in the Single Image setting.

| Image resolution | SIFID ↓ | LPIPS ↑ |
|---|---|---|
| 192x320 | 0.08 | 0.33 |
| 512x896 | 0.08 | 0.29 |

image resolution of 896x512. We show the visual results for high resolution image synthesis of SIV-GAN in Fig. J. We don't observe any issues caused by the change of image resolution. As shown in Table I, the performance of the model is similar at different scales: SIFID of 0.08 and LPIPS of 0.29 at resolution 512x896 is aligned well with quality and diversity at resolution 192x320 (0.08 and 0.33 in Table 1).

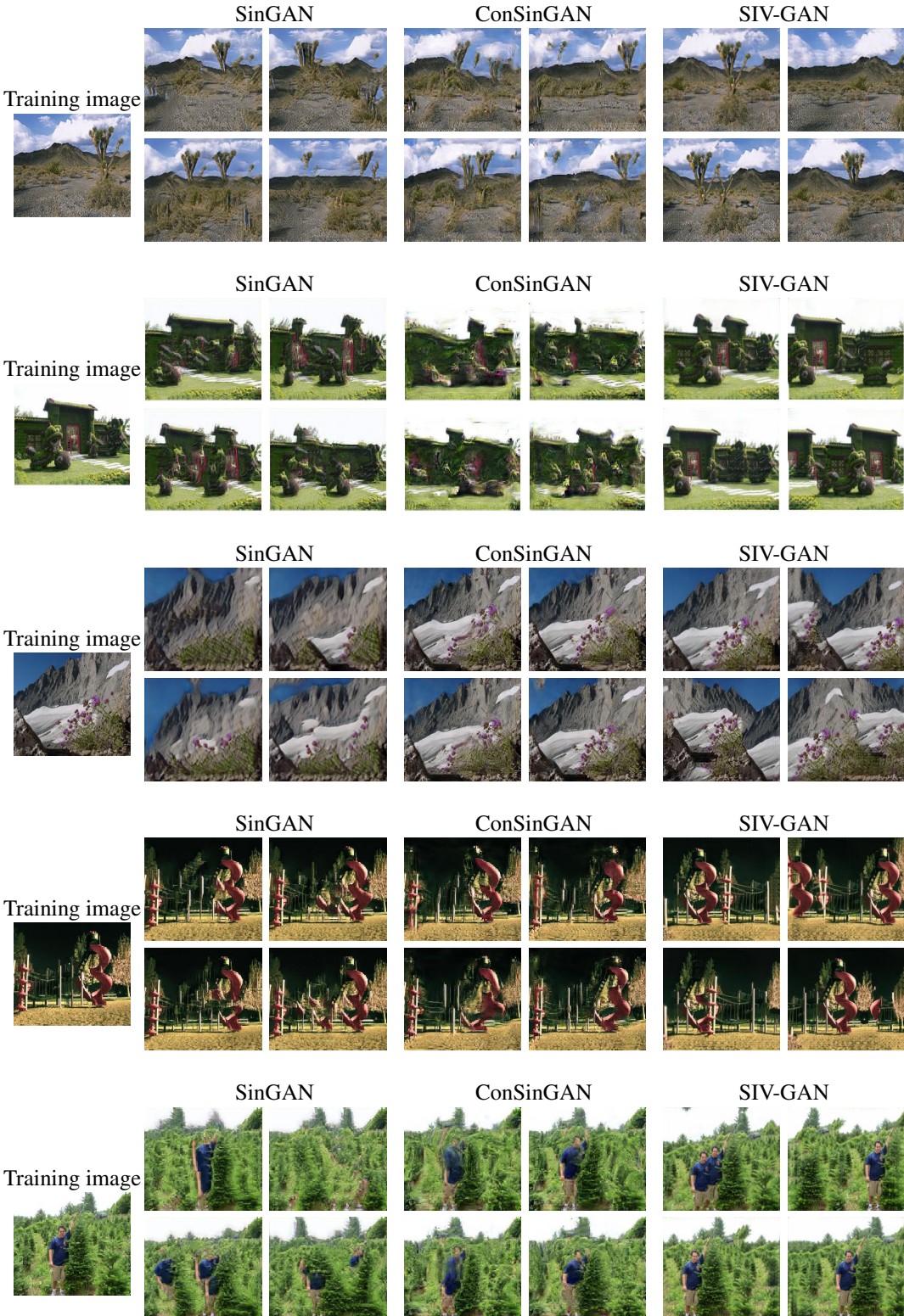

Figure G: Comparison with other methods in the Single Image setting on Places. Single-image GANs of (Shaham et al., 2019; Hinz et al., 2021) are prone to incoherently shuffle image patches. In contrast, SIV-GAN produces images preserving the appearance of objects.

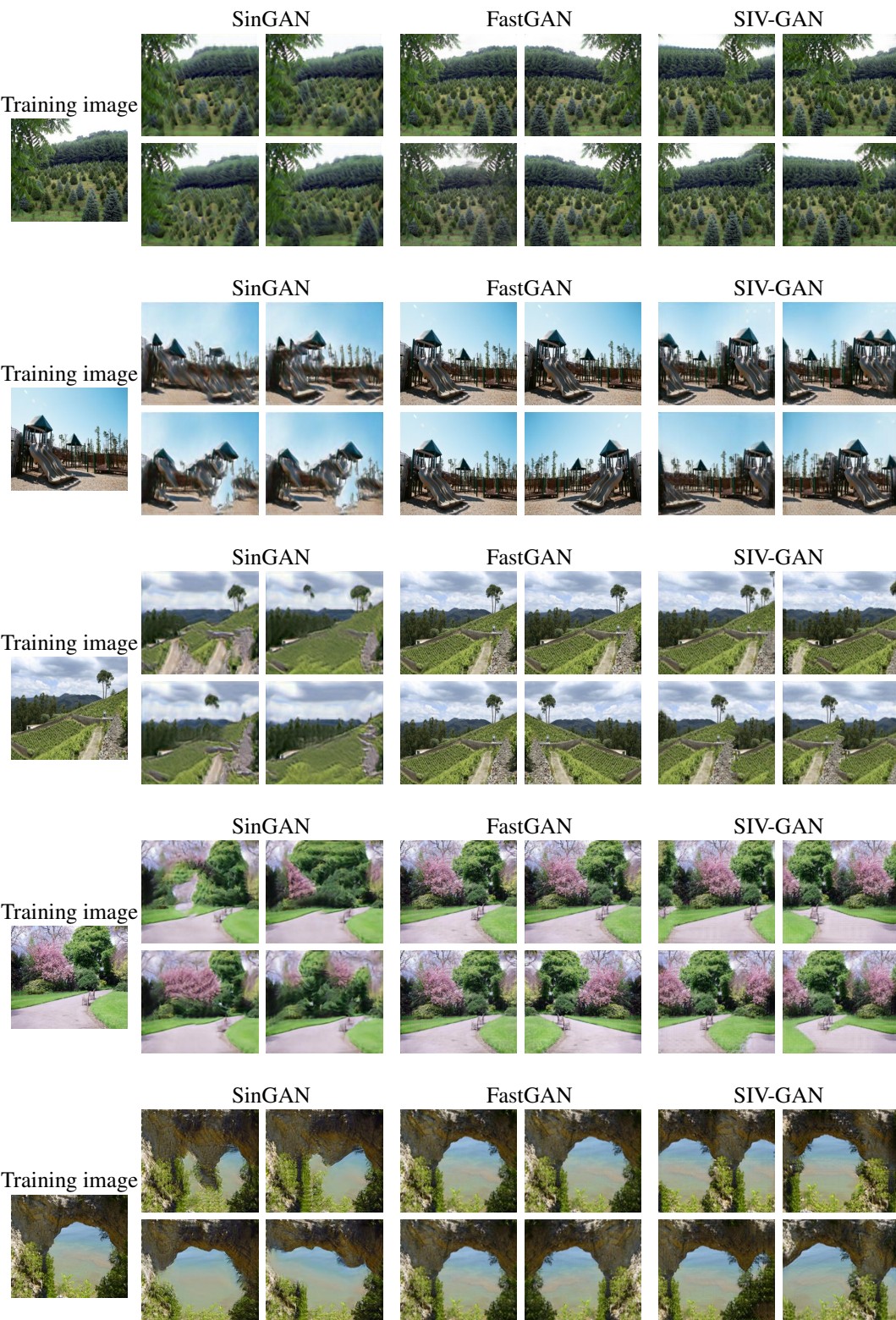

Figure H: Comparison with other methods in the Single Image setting on Places. The single-image GAN of (Shaham et al., 2019) is prone to incoherently shuffle image patches, and the few-shot FastGAN model (Liu et al., 2021) reproduces the training image or its flipped version. In contrast, SIV-GAN produces diverse images, preserving the appearance of objects.

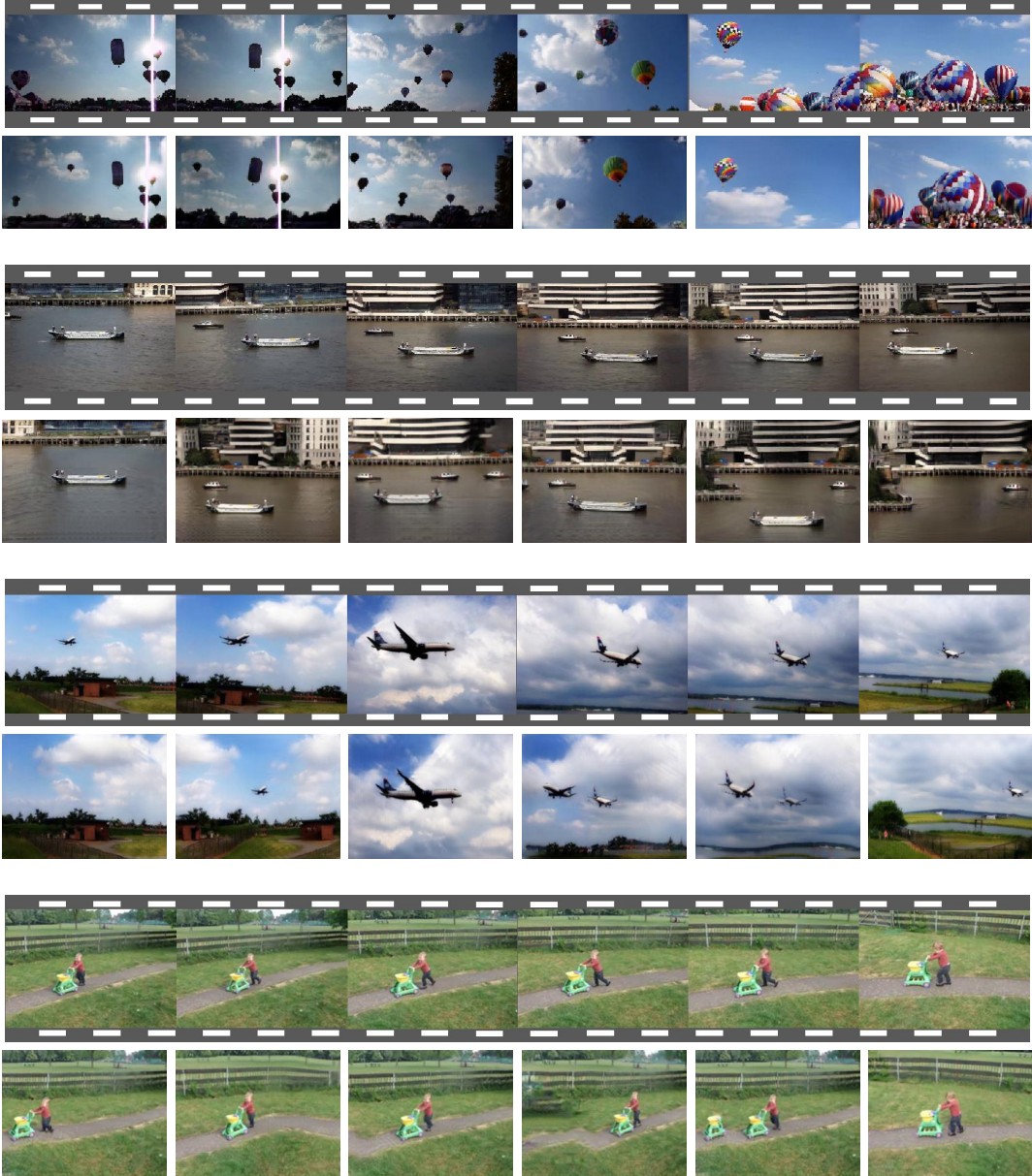

Figure I: Additional qualitative results in the Single Video setting. The training sequences are shown in grey frames. Given a single video for training, our SIV-GAN produces images that are different to the training frames. For example, for a video with air balloons, the generated images have different number of balloons, while for a video with a boat, the model generates novel combinations of boats and buildings not seen during training.

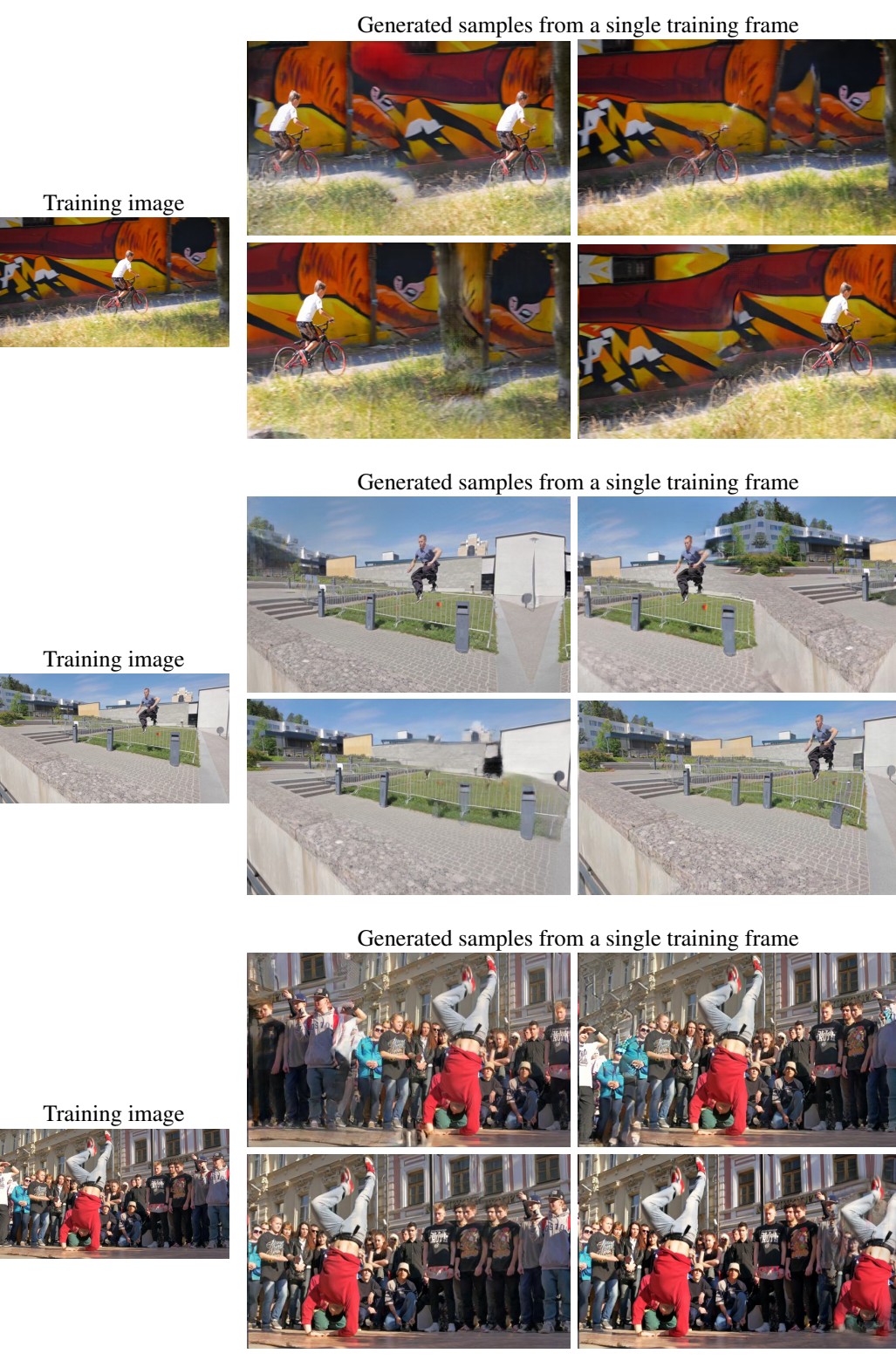

Figure J: SIV-GAN results at a high image resolution of 512x896 in the Single Image setting. Our model shows good scalability to different resolutions, maintaining the quality and diversity of scene compositions. For example, given only one image with a bike rider, SIV-GAN can produce image with two bikers, without the biker, as well as change the bike lane path. For the frame with a parkour jumper, the scene can be resynthesized without the jumper, or with a different positioning of barriers. Finally, for the image with a breakdancer, the number and positions of the dancer and observers can be changed.

### D.3 SIFID USING DIFFERENT INCEPTIONV3 LAYERS

To evaluate image quality at different scales, we additionally compute the SIFID metric (Shaham et al., 2019) at various InceptionV3 layers. Its original formulation uses InceptionV3 features before the first pooling layer, at a spatial resolution of $\frac{H \times W}{4}$. Such metric captures only low-level image details, such as colors and textures, and is not able to capture high-level semantic image properties, such as appearance of objects or global layouts. Therefore, to evaluate realism at different scales, we additionally use features before the second pooling layer (at a resolution of $\frac{H \times W}{8}$), pre-classifier features ($\frac{H \times W}{8}$) and the final Inception features ($1 \times 1$), that are commonly used to compute FID (Heusel et al., 2017). To obtain the SIFID score in the Single Image setting, we generate 100 images and then compute their mean SIFID to the training image (corresponding to columns *mean* in Table 1). The results on the DAVIS-YFCC100M dataset (Perazzi et al., 2016; Thomee et al., 2016) for the Single Image and Single Video settings are presented in Table J.

We observe no disagreement with the metrics reported in the main paper. SIV-GAN achieves better image quality than the comparison models in both settings, as measured at all the InceptionV3 layers. Notably, Table J shows that single-image methods, SinGAN and ConSinGAN, achieve comparable low-level SIFID (at a resolution of $\frac{H \times W}{4}$), but achieve poor high-level scores ($\frac{H \times W}{16}$, $1 \times 1$). This indicates that these models successfully learn textures of given images, but fail to reproduce scenes at a mid-level (objects) and the global scale (layout). In contrast, our proposed two-branch discriminator allows learning the scene appearance at all scales, enabling generation of plausible images not only with fine textures, but also with correct appearance of objects and globally-coherent layouts.

Table J: Comparison of SIFID at different scales on DAVIS-YFCC100M in the Single Image and Single Video settings.

| | Single Image | | | | Single Video | | | |
|---|---|---|---|---|---|---|---|---|
| | $\frac{H \times W}{4}$ | $\frac{H \times W}{8}$ | $\frac{H \times W}{16}$ | $1 \times 1$ | $\frac{H \times W}{4}$ | $\frac{H \times W}{8}$ | $\frac{H \times W}{16}$ | $1 \times 1$ |
| SinGAN | 0.13 | 4.93 | 34.52 | 2510 | 2.47 | 13.61 | 96.35 | 411 |
| ConSinGAN | 0.09 | 2.94 | 27.33 | 1960 | 2.74 | 14.73 | 74.50 | 392 |
| FastGAN | 0.13 | 2.89 | 19.48 | 1340 | 0.79 | 1.75 | 9.24 | 141 |
| SIV-GAN | **0.08** | **1.29** | **16.30** | **1100** | **0.55** | **1.32** | **5.14** | **115** |

### D.4 COMPARISON OF SYNTHESIS DIVERSITY IN THE SINGLE IMAGE AND SINGLE VIDEO SETTINGS

In Table K we compare the diversity among the images generated from a single image and from a single video on the DAVIS-YFCC100M dataset. In the Single Image setting we use only one frame in the middle of the sequence for training, while in the Single Video setting we use all video frames as training data. As seen from the table, the diversity of generated samples is notably higher for the case when all frames of a video are used (0.43 against 0.33 LPIPS). The visual difference between the settings is illustrated in Fig. K. The model, trained only on the middle frame, produces slight variations of the training image. For example, such model changes the number of windows in the building or modifies the geometry of the concrete barrier. On the other hand, the model, trained on a full video, achieves more complex transformations, being capable of changing the layout of buildings or removing a person from the scene.

In practice, capturing a short video clip can take almost as little effort as collecting an image. As we show in our experiments, using short videos for training allows to generate significantly more diverse images, compared to the case when only a single image is used. This way, we would like to

Table K: Comparison of the diversity among the images generated in the Single Image and the Single Video settings on the DAVIS-YFCC100M dataset.

| Data setting | LPIPS ↑ | MS-SSIM ↓ |
|---|---|---|
| Single Image | 0.33 | 0.63 |
| Single Video | **0.43** | **0.54** |

Generated images from the single frame

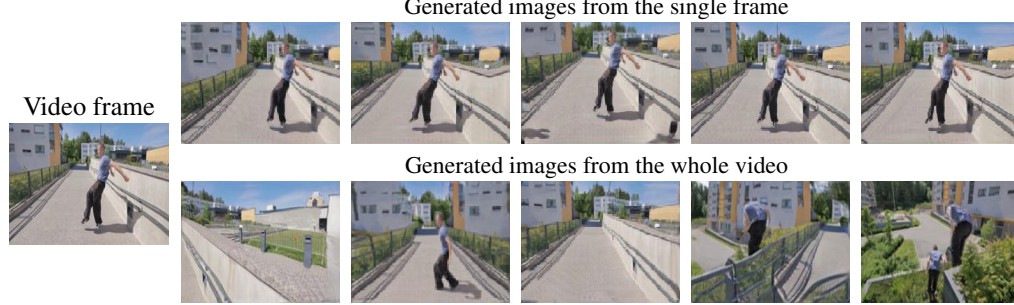

Video frame

Generated images from the whole video

Figure K: Difference between models trained on a single middle frame of a video and on the full video sequence. In the Single Image setting, the model generates only slight geometric transformations, such as varying the number of windows on a building, or modifying the geometry of a concrete barrier. In contrast, the Single Video setting allows to provide more substantial diversity, removing the person from the frame or changing the layout of buildings.

draw the attention of the community to the Single Video setting, which we believe to be helpful in extending the usability of generative models for practical applications.

### D.5 QUALITATIVE EFFECT OF THE PROPOSED DIVERSITY REGULARIZATION AND FEATURE AUGMENTATION

The proposed diversity regularization (DR) and feature augmentation (FA) are essential components for SIV-GAN to achieve a high diversity of synthesis. Table 3 demonstrates a quantitative effect of these two components on the performance of our model. In Fig. L we supplement this analysis with a visual study, showing the images generated by our full model, as well as by the models trained without DR or FA in the Single Image setting.

We observe that the visual results correspond well to the numbers from Table 3. As seen from Fig. L, SIV-GAN without DR does not manage to mitigate overfitting, suffering from mode collapse. Such a model learns to memorize the original training sample and thus reproduces it without any modifications. In Table 3 this corresponds to the low LPIPS and Dist. to train scores of 0.04 and 0.06. The model with DR but without FA manages to achieve diverse image synthesis. However, such model produces only modest diversity in content and layouts. For example, for an image with waves and surfers, it does not change the number of surfers, while for images with rocks, it typically translates rocks to new locations but does not change their shape. In Table 3 this is reflected in the increased LPIPS and Dist. to train scores of 0.27 and 0.33. Finally, our full model, trained with both DR and FA, enables generating more interesting novel scene compositions, varying global scene layouts and changing the content distribution. For instance, it can generate different number of surfers or mountains compared to the training sample, or modify the shapes of rocks. Correspondingly, in Table 3 our full model shows a much higher diversity, achieving the highest LPIPS of 0.33 and Dist. to train of 0.37.

## E DETAILED DESCRIPTION OF THE ARCHITECTURE AND TRAINING DETAILS

### E.1 ANALYSIS OF STRATEGIES TO GENERATE IMAGES OF RECTANGULAR SHAPES

Commonly used state-of-the-art GAN models, such as (Brock et al., 2019; Karnewar & Wang, 2020), generate images of square shapes. In our studied settings, single images or video frames can be rectangular, having resolutions of different horizontal and vertical aspect ratios. In contrast, we design our model to be compatible with images of different resolutions. In this section, we analyse two possible strategies to achieve this.

One possible strategy is to follow the design of SinGAN (Shaham et al., 2019), which has several training stages. Such model proposes to reach the final resolution following a geometric progression, where the image is upsampled after each training stage. In this scheme, the progression multiplier is computed based on the image size, the number of stages, and a predefined image size at first model stage. Typically, this multiplier is non-integer and is different for horizontal and vertical axes. For SIV-GAN, we adopted such scheme by progressively upsampling the image after each generator

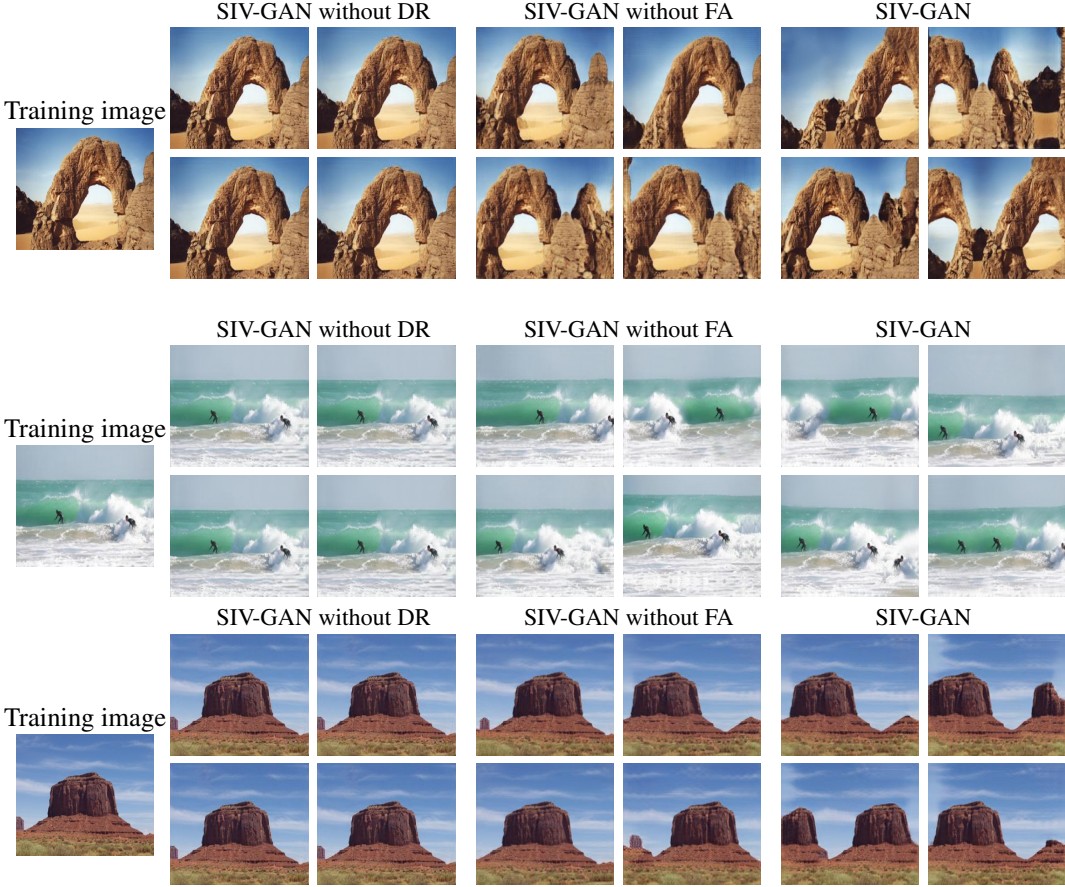

Figure L: Qualitative ablation on the proposed diversity regularization and feature augmentation. Without DR, the model does not mitigate memorization, producing only images that are perceptually indistinguishable from the original sample. Without FA, SIV-GAN achieves only modest diversity in content and layouts. Finally, using both DR and FA enables generating more interesting novel scene compositions, varying global scene layouts, or duplicating and removing objects.

block, starting from noise of shape $(4 \times 4)$ and reaching the exact image shape by the final generator layers. In our experiments, such upsampling scheme leads to texture artefacts in generated images (see Fig. M). We observed that the artefacts are caused by non-integer interpolation between layers: backpropagation through such transformation causes a "grid" effect, which is hard to correct during training of the generator. The difference in the texture quality in comparison to Shaham et al. (2019); Hinz et al. (2021) is explained by the fact that these models are trained in multiple stages, without requiring backpropagation through upsampling layers. In contrast, our SIV-GAN is a single stage model, and gradients cannot bypass the interpolation.

To avoid corruptions of textures, we therefore select an alternative solution. As discussed in App. E.2, we propose to follow the architecture of Brock et al. (2019), employing x2 upsampling after each residual block, but modifying the shape of input noise, adjusting it to fit better to the final image

Table L: Effect of different strategies to match the rectangular shape of the original training image on the DAVIS-YFCC100M dataset.

| Interpolation | Single Image | | Single Video | |
|---|---|---|---|---|
| | SIFID ↓ | LPIPS ↑ | SIFID ↓ | LPIPS ↑ |
| Progressive upsampling | 0.11 | 0.32 | 0.65 | 0.42 |
| Adjusted noise | **0.08** | **0.33** | **0.55** | **0.43** |

resolution. Note that for this approach the final resolution of generated images is an even multiple of the input noise shape, which can sometimes not precisely fit to the resolution of an original image (for example, in case when the training image has an odd number of pixels in one of the dimensions). Nevertheless, with such approach the shape of an image is preserved better, compared to generating square images as in (Brock et al., 2019; Karnewar & Wang, 2020). As seen from Fig. M, adjusting the noise shape leads to better quality of images, overcoming the interpolation problem.

Table L confirms our analysis. We observe that using the progressive upsampling, as in Shaham et al. (2019), leads to decreased quality of images, as measured by SIFID. Moreover, such approach produces slightly lower diversity. Therefore, for our final model design, we choose to adjust input noise shape, as it leads to better performance.

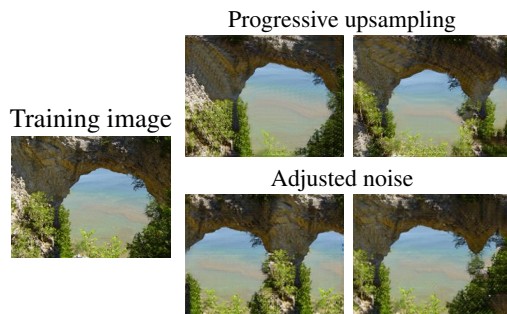

### E.2 ARCHITECTURE AND TRAINING DETAILS

Architecture details of our SIV-GAN are summarized in Tables M and N. Our model has a generator $G$ and a discriminator $D$ consisting of three parts: low-level feature extractor $D_{low-level}$, layout branch $D_{layout}$, and content branch $D_{content}$. Our design mainly follows the one of BigGAN (Brock et al., 2019), applying ResNet blocks at both the generator and the discriminator. Following MSG-GAN

Figure M: Different strategies to achieve rectangular shapes. Using progressive upsampling, as in (Shaham et al., 2019), leads to artifacts in textures (e.g. chequered "waves" on the rock). Controlling output image resolution through adjusting noise shape helps to overcome this issue and generate textures of good quality.

(Karnewar & Wang, 2020), we use skip-connections between $G$ and $D$. We thus generate images at different resolutions and pass them to discriminator at different scales. In contrast to (Karnewar & Wang, 2020), our discriminator is not symmetric to generator, as after $N_{low-level}$ discriminator blocks we apply branching. Therefore, we generate images only at $N_{low-level}$ highest resolutions, and then pass them at each block of $D_{low-level}$. To incorporate generated images at intermediate $D$ blocks, we apply a $1 \times 1$ convolution, and concatenate the obtained tensor to the current $D$ features (strategy $\phi_{lin\_cat}$ from (Karnewar & Wang, 2020).)

The branches $D_{layout}$ and $D_{content}$ are formed by usual $D$ ResNet blocks, with two modifications. First, as the content branch has $1 \times 1$ spatial dimensions, the used convolutions have kernel size $(1 \times 1)$ and are technically equivalent to a fully-connected layer. Secondly, the layout branch has 1 channel, so its convolutions always have the number of channels equal to 1. To form a binary real-fake decision tensor, after each $D$ block we use a $1 \times 1$ convolution, which maps the current features to a one-channel map with logits.

In order to fit better to the training image size, we change the resolution of the input noise. We keep the noise resolution around $(4 \times 4)$, adjusting it to different ratios to fit shapes of training images closer. For example, the input noise in our experiments was resized to $(3 \times 4), (3 \times 5)$ or $(4 \times 4)$, depending on the final image resolution. For evaluation, we bilinearly resized generated images to the resolution of original training image. Tables M and N present an example for input noise of resolution $3 \times 5$, which correspond to the frames from the DAVIS video dataset (Perazzi et al., 2016).

We train our model with ADAM (Kingma & Ba, 2015) optimizer, using a batch size of 5, momentums $(\beta_1, \beta_2) = (0.5, 0.999)$, learning rates $0.0002$ for both the generator and the discriminator. All our experiments were conducted on a single GTX 1080 GPU with 12 GB RAM memory.

Table M: The SIV-GAN generator. In this example, the configuration is presented for the input noise of size $(3 \times 5)$ and the final resolution of $(192 \times 320)$, corresponding to training on the DAVIS-YFCC100M dataset in the Single Video setting.

| Operation | Input | Size | Output | Size |
|---|---|---|---|---|
| ConvTransp2D | z | (64,1,1) | up_0 | (256,3,5) |
| ResBlock-Up | up_0 | (256,3,5) | up_1 | (256,3,5) |
| ResBlock-Up | up_1 | (256,3,5) | up_2 | (256,6,10) |
| ResBlock-Up | up_2 | (256,6,10) | up_2 | (256,12,20) |
| ResBlock-Up | up_3 | (256,12,20) | up_3 | (256,24,40) |
| ResBlock-Up | up_4 | (256,24,40) | up_4 | (256,48,80) |
| ResBlock-Up | up_5 | (256,48,80) | up_5 | (128,96,160) |
| ResBlock-Up | up_6 | (128,96,160) | up_6 | (64,192,320) |
| Conv2D, TanH | up_5 | (128,48,80) | image_2 | (3,48,80) |
| Conv2D, TanH | up_6 | (64,96,160) | image_1 | (3,96,160) |
| Conv2D, TanH | up_7 | (32,192,320) | image_0 | (3,192,320) |

Table N: The SIV-GAN discriminator. In this example, the configuration is presented for the input noise of size $(3 \times 5)$ and the final resolution of $(192 \times 320)$, corresponding to training on the DAVIS-YFCC100M dataset in the Single Video setting.

| Operation | Input | Size | Output | Size |
|---|---|---|---|---|
| | Low-level features $D_{low-level}$ | | | |
| Conv2D | image_0 | (3,192,320) | feat_0 | (32,192,320) |
| Conv2D | image_1 | (3,96,160) | feat_1 | (8,96,160) |
| Conv2D | image_2 | (3,48,80) | feat_2 | (16,48,80) |
| ResBlock-Down | feat_0 | (32,192,320) | down_0 | (64,96,160) |
| ResBlock-Down | down_0 | (64,96,160) | down_1 | (128,48,80) |
| | feat_1 | (8, 96, 160) | | |
| ResBlock-Down | down_1 | (128,48,80) | F | (256,24,40) |
| | feat_2 | (16,48, 80) | | |
| | Content branch $D_{content}$ | | | |
| AvgPool | F | (256,24,40) | F_con | (256,1,1) |
| ResBlock-Down | F_con | (256,1,1) | cont_0 | (256,1,1) |
| ResBlock-Down | cont_0 | (256,1,1) | cont_1 | (256,1,1) |
| ResBlock-Down | cont_1 | (256,1,1) | cont_2 | (256,1,1) |
| ResBlock-Down | cont_2 | (256,1,1) | cont_3 | (256,1,1) |
| | Layout branch $D_{layout}$ | | | |
| Conv2D | F | (256,24,40) | F_lay | (1,24,40) |
| ResBlock-Down | F_lay | (1,24,40) | lay_0 | (1,12,20) |
| ResBlock-Down | lay_0 | (1,12,20) | lay_1 | (1,6,10) |
| ResBlock-Down | lay_1 | (1,6,10) | lay_2 | (1,3,5) |
| ResBlock-Down | lay_2 | (1,3,5) | lay_3 | (1,3,5) |

