# OpenReview forum: "Generating Novel Scene Compositions from Single Images and Videos"
_ICLR.cc/2022/Conference — ICLR 2022 Submitted_

### Official Review · Reviewer_LNdN · 2021-11-02

**Correctness:** 3
**Technical Novelty And Significance:** 3
**Empirical Novelty And Significance:** 2
**Recommendation:** 5
**Confidence:** 5

**Main Review:**

This paper introduces a new GAN architecture for unconditional image synthesis after being trained on a single image/video. Based on the observation that previous models either fail to generate realistic layouts (SinGAN, ConSinGAN) or collapse to the training data without diversity (FastGAN) the paper proposes a new discriminator architecture to address this.
After a couple of layers to extract image features, the discriminator splits into two independent branches. One branch is trained to judge the global image layout while the second branch is trained to evaluate individual object quality. Aditionally, a feature augmentation is introduced to further increase the diversity.
At the generator side, a diversity regularization is added which forces to generator to generate different images for different latent samples.

The resulting model outperforms all baselines when comparing the image quality and diversity. Compared to single-image GANs it obtains a much better quality by generating realistic layouts and compared to few-shot GANs it obtains a much higher diversity. A detailed ablation study shows the effects and improvements of the individual novelties.

While the approach is intuitive and does seem to lead to clear improvements I wonder how well it translates to applications such as image harmonization/editing/animation/retargeting/etc. While unconditional synthesis is interesting I think many of the more obvious direct applications do not involve pure unconditional generation. Since at least the single-image GANs seem to be able to perform those tasks without or only minor changes in the training pipeline it should be relatively simple to evaluate this approach on some of these tasks, too.

Related to that: while the diversity regularization in general makes sense I wonder if really forcing the features to be different regardless of the distance in z is the best way of doing. This may affect tasks such as animation or latent space interpolation. Have you tried the "normal" way of just forcing the features to be different dependent on the distance in z? I don't see a compelling reason why this shouldn't work, too.

Also, it would be helpful to have a user study to compare your approach with the baselines. Since the main metrics (to a degree) oppose each other (e.g. SIFID vs diversity), having humans compare the models directly would be helpful. You could either reproduce the user studies from SinGAN/ConSinGAN (compare against training image) or compare the models against each other directly.

Finally, do you have any numbers on the model size compared to the baselines, training time/effort, etc? It seems like your model is trained for many more iterations than SinGAN/ConSinGAN and not in a progressive manner, i.e. I assume it takes longer to train your model?



**Summary Of The Paper:**

This paper proposes a single-image/single-vide GAN model that obtains both high quality and high diversity in the generated images, as opposed to previous models that only achieve one of the two. To achieve this, the discriminator consists of two branches that evaluate global layout and quality independently and a novel regularization approach for the generator is introduced. Comparisons to SinGAN, ConSinGAN, and FastGAN show improved performance.

**Summary Of The Review:**

The proposed novleties make sense and seem to lead to clear improvements. However, the evaluation is only based on unconditional image synthesis and ignores many more practical applications for single-image models such as animation, harmonization, retargeting, etc. A more broad evaluation on some of these tasks coupled with a user study would be more appropriate.

---

> ### Author Response · Authors · 2021-11-17
> **Response to Reviewer LNdN**
>
> **LNdN-3:** *“Also, it would be helpful to have a user study to compare your approach with the baselines. Since the main metrics (to a degree) oppose each other (e.g. SIFID vs diversity), having humans compare the models directly would be helpful. You could either reproduce the user studies from SinGAN/ConSinGAN (compare against training image) or compare the models against each other directly.”*
>
> We thank you for the suggestion. We will attach our findings from the user study as a separate comment to this post. In the meantime, let us take an opportunity to share a perspective on why our quantitative evaluation may provide a more adequate comparison between the models compared to a human study from previous works.
>
> Aside from being subjective and hardly reproducible, human studies do not broadly evaluate both the quality and diversity of image synthesis [4]. For example, a qualitative user assessment based on samples is very likely to be biased towards models which overfit. Therefore, in our extremely low data regimes, such evaluation would in most cases disfavor significant deviations from the original training data. This generally conflicts with our motivation, as the model selected by the human perception may have less diversity, and thus provide fewer outliers which can be useful for data augmentation. In contrast, our evaluation in Sec. 4 separately assesses the image quality (SIFID), the perceptual (LPIPS) and pixel diversity, as well as the average distance to training data (Dist. to train). Therefore, our evaluation provides a much broader insight into the performance of the models.
>
> Further, as humans are better at perceiving images at a high object level, their evaluation downweighs the image realism at other image levels. We argue that this bias should be addressed by the quantitative analysis. Particularly, in Appendix D.3 we report SIFID at different InceptionV3 layers, corresponding to image realism at different layers. Interestingly, SIFID at different layers can provide a different ranking of the models, indicating that a fair analysis should also include quantitative inspection at different scales.
>
> [4] [Pros and Cons of GAN Evaluation Measures.](https://arxiv.org/pdf/1802.03446.pdf) Ali Borji, Arvix
>
> $~$
>
> **LNdN-4:** *“Finally, do you have any numbers on the model size compared to the baselines, training time/effort, etc? It seems like your model is trained for many more iterations than SinGAN/ConSinGAN and not in a progressive manner, i.e. I assume it takes longer to train your model?”*
>
> Please see our Table below, where we provide the number of parameters and training time of the models. All the numbers correspond to training on a single image taken from DAVIS-YFCC100M, at the image resolution of 320x192. Following [5], we report the parameter count only for the generator network which is used at inference. We use the default parameters for all the models, and train the few-shot FastGAN model for 100k iterations, like our model.
>
> As also discussed in *LNdN-1*, we refrain from using a multi-stage training scheme. The one-stage structure of our model makes it more similar to standard GAN approaches, i.e., increases the training time, but on the other hand enables its application to multiple images. Therefore, it is more fair to compare SIV-GAN to single-stage GANs that also support multiple images. As seen in the Table, our training time is comparable to the FastGAN model, which was introduced as a lightweight few-shot model with a small computational cost [6]. Meanwhile, the parameter count for our model is significantly lower.
>
> | Model  | Supports multiple images? | Parameters | Time |
> |---|---|---|---|
> | SinGAN |  | 1.3M | 2.5 h |
> | ConSinGAN |  | 0.6M | 30 min |
> | SIV-GAN | ✓ | 6M | 7 h |
> | FastGAN | ✓ | 29M | 7 h |
>
> [5] [Improved Techniques for Training Single-Image GANs.](https://arxiv.org/abs/2003.11512) Hinz et al, WACW21
> [6] [Towards Faster and Stabilized GAN Training for High-fidelity Few-shot Image Synthesis.]( https://arxiv.org/abs/2101.04775) Liu et al, ICLR21
>
> $~$
>
> In conclusion, we hope that our answers were helpful to clarify the details of our work, particularly in regards to the difference of our setup and motivation to the one of prior single-image works [5,6]. We are looking forward to hearing your feedback!

---

> > ### Author Response · Authors · 2021-11-18
> > **User study**
> >
> > **LNdN-3(1):**
> >
> > Following your suggestion, we conduct a human study in which we ask users to compare the perceptual quality of generated images. To avoid biased opinions towards our model, we invited the participants not from our lab, and who were not familiar with our work. Following the protocol of SinGAN and ConSinGAN [1,2], the workers were proposed two types of polls:
> > -	*Confusion vs real.* Workers were presented a sequence of trials, in which a randomly generated image was shown side by side with the corresponding training image. Each pair was shown for 1 second. The participants were asked to pick the fake image.
> > -	*User preference.* In each trial, we showed a random sample generated by SinGAN, ConSinGAN, and SIV-GAN. The original training image was not shown, and we did not introduce a time limit. The workers were asked to pick the most realistic image among fake samples.
> >
> > The study was conducted on the Places dataset, resulting in 50 comparisons for each experiment. For each experiment the images were sampled randomly, so each participant was provided with a different set of images.
> >
> > The results are shown in the Table below. We observe a clear preference of our images among the participants. In a *user preference* study, our images were chosen as most realistic in 44.4% of cases, with the closest competitor SinGAN gaining 31.2%.  Our images were also more frequently taken for real in the *confusion* study, fooling the users in almost every third trial.
> >
> > We hypothesize that the success of SIV-GAN in the user preference study is mainly related to its ability to preserve the appearance of *large-* and *middle-* size objects. While the patch-based methods SinGAN and ConSinGAN frequently distort objects by permuting their patches, our model offers the content and layout diversity at a more global scale, and thus preserves the realism of objects better (as can be also seen in Fig G, where we provide a qualitative comparison on Places).
> >
> > | Model | User preference $\uparrow$ | Confusion vs real $\uparrow$ |
> > |---|---|---|
> > | SinGAN | 31.2% ± 2.0% | 24.0% ± 2.2% |
> > | ConSinGAN | 24.4% ± 2.7% | 20.0% ± 2.5% |
> > | SIV-GAN | **44.4% ± 3.2%** | **30.4% ± 1.5%** |
> >
> > $~$
> >
> > [1] [SinGAN: Learning a Generative Model from a Single Natural Image.]( https://arxiv.org/abs/1905.01164) Shaham et al, ICCV19
> > [2] [Improved Techniques for Training Single-Image GANs.](https://arxiv.org/abs/2003.11512) Hinz et al, WACV21

---

> > > ### Comment · Reviewer_LNdN · 2021-11-29
> > > **Thanks for the user study**
> > >
> > > I appreciate the authors' user study which shows clear preferences for their model compared to the baselines. However, I still think that evaluating on only unconditional image synthesis is not enough at this point though, especially at a train time of 7h per image. While I understand that the authors see this differently I still think evaluating on more tasks is necessary to make this approach convincing and I would, therefore, stay with my original rating.

---

> > > > ### Author Response · Authors · 2021-11-29
> > > > **Reply to LNdN**
> > > >
> > > > Thank you for appreciating our user study. However, we could not agree that having only unconditional evaluation makes our approach unconvincing. Firstly, as outlined in Sec. 2 and answer *LNdN-1*, our approach is technically closer to the line of few-shot unconditional works [1,2,3]. Such approaches, dealing with several images, have different requirements on training time. We also do not claim any improvement on conditional settings and thus find it unfair to ask for comparisons with purely single-image methods (which are conditional through their multi-stage architectures) on conditional tasks.
> > > >
> > > > Secondly, the applications of GANs are not limited to conditional scenarios. Unconditional image sampling also has interesting applications. For example, in many practical domains, it can serve as data augmentation, which is especially useful in low-data scenarios (see answer *LNdN-1* and *DXh5-2*).
> > > >
> > > > Finally, it is standard practice to evaluate models only on unconditional low-data settings [1,2,3]. We find that our contribution to this task is convincing, because it offers better performance than previous models, at the same time being more universal. It can be applied across very different unconditional data regimes: *single images*, *single videos* (60-100 very similar images), as well as to *few-shot datasets* (100-389 images, Appendix. C). Note that this was not possible with previous one-shot and few-shot GAN models (see Sec. 4).
> > > >
> > > > [1] [Towards Faster and Stabilized GAN Training for High-fidelity Few-shot Image Synthesis](https://arxiv.org/abs/2101.04775), Liu et al, ICLR21
> > > > [2] [Differentiable augmentation for data-efficient gan training](https://arxiv.org/pdf/2006.10738.pdf). Zhao et al, NeurIPS20
> > > > [3] [Regularizing Generative Adversarial Networks under Limited Data](https://arxiv.org/abs/2104.03310), Tseng et al, CVPR21

---

> ### Author Response · Authors · 2021-11-17
> **Response to Reviewer LNdN**
>
> **LNdN-2:** *“Related to that: while the diversity regularization in general makes sense I wonder if really forcing the features to be different regardless of the distance in z is the best way of doing. This may affect tasks such as animation or latent space interpolation. Have you tried the "normal" way of just forcing the features to be different dependent on the distance in z? I don't see a compelling reason why this shouldn't work, too.”*
>
> Similar to our response *LNdN-1*, we would elaborate that our goal is more concerned with the *diversity of synthesis*, rather than achieving smooth transitions in the latent space. Considering a potential application to data augmentation, our design choices promote sample diversity, their high distances to training data, while also aim to maintain their semantic context.
>
> Using an alternative solution to regularize the difference of images based on the distance between their latent codes, e.g., $(G(z1)−G(z2))/(z1−z2)$, is certainly possible. This is known as a Diversity Sensitive loss (DS) [3]. We have tried to apply the DS to our model, but have unfortunately observed only modest diversity with it. This is demonstrated in Table 4, where our proposed DR (also applied in the image space) significantly outperforms DS, improving LPIPS from $0.14$ to $0.21$. Therefore, omitting the denominator to make the regularization more aggressive, encouraging the images to be different even for very close latent codes $z1$ and $z2$, was a very beneficial step to achieve high diversity.
>
> Note that [3] studied GAN training on sufficiently large datasets, which is a different setup from ours. In our case of a single image or a single video, the whole latent space of the generator should model only one semantic scene. As the latent space contains no transitions between semantic domains, the perceptual difference between images should be much less correlated to their distance in the latent space. Based on the above analysis and considering our goal, in our settings it is more meaningful to apply the regularization uniformly to all pairs of generated samples.
>
> [3] [Diversity-sensitive conditional generative adversarial networks.](https://arxiv.org/abs/1901.09024) Yang et al

---

> ### Author Response · Authors · 2021-11-17
> **Response to Reviewer LNdN**
>
> Thank you for your feedback! We would like to take the opportunity to share our perspective on the questions you have raised:
>
> $~$
>
> **LNdN-1:** *“While the approach is intuitive and does seem to lead to clear improvements I wonder how well it translates to applications such as image harmonization/editing/animation/retargeting/etc. While unconditional synthesis is interesting I think many of the more obvious direct applications do not involve pure unconditional generation. Since at least the single-image GANs seem to be able to perform those tasks without or only minor changes in the training pipeline it should be relatively simple to evaluate this approach on some of these tasks, too.”*
>
> We agree with you that conditional single image synthesis has lots of interesting applications. Yet, we kindly point out that the focus of our paper is significantly different.
>
> We recall that our overall goal is to mitigate the memorization issue of unconditional GANs trained on extremely small datasets. Thus, rather than enabling editing applications, we specifically target the *diversity of synthesis*, aiming to generate structurally novel scene compositions. Note that SIV-GAN is indeed more successful than SinGAN and ConSinGAN in mitigating memorization, achieving much higher LPIPS diversity and distance to training data In Table 1. Moreover, our model is not restricted to single images, also being capable of learning from multiple images. For example, our model successfully learns from multiple frames of a single video (Fig. 5, Table 2), and can also be applied in the few-shot image generation setting (see Appendix C). Therefore, our model design is motivated by a different scope of applications.
>
> Remarkably, diverse unconditional synthesis from limited data also has a wide range of applications. In many practical one-shot and few-shot scenarios, a lack of data is a bottleneck to achieving good performance. A common approach to avoid overfitting under a critical lack of data in such tasks is to diversify the available data by employing image augmentation. Albeit helpful, standard image augmentation techniques, such as rotation or flipping, provide only limited diversity, for example always preserving relative positions of objects in images. The synthetic augmentation from our model overcomes this limitation. As seen in Figure 1, SIV-GAN can put objects at different locations, swap their positions, or even duplicate or remove some of them, which is not achievable with standard data augmentation techniques. Therefore, we manage to produce image augmentation of higher structural diversity, which can provide additional gains when utilized as extra data augmentation in downstream applications. We additionally refer you to our answer to another reviewer *DXh5-2* for possible examples of such applications.
>
> Motivated by *unconditional* synthesis in both *single-* and *multi-* image regimes, our model is conditioned only on input noise and is not designed for any other source of conditional information. However, like other unconditional GAN models, SIV-GAN can support image editing via latent space exploration [1]. In such a pipeline, an image of interest can be embedded into the generator’s latent space [2], and then the manipulation can be performed via modifying the latent vector. We consider this as an interesting direction for future work.
>
> [1] [Image2StyleGAN: How to Embed Images Into the StyleGAN Latent Space?](https://arxiv.org/abs/1904.03189) Abdal et al, ICCV19
> [2] [Unsupervised Discovery of Interpretable Directions in the GAN Latent Space.](https://arxiv.org/abs/2002.03754) Voynov et al, ICML20

---

### Official Review · Reviewer_DXh5 · 2021-11-02

**Correctness:** 3
**Technical Novelty And Significance:** 2
**Empirical Novelty And Significance:** 2
**Recommendation:** 5
**Confidence:** 4

**Main Review:**


##########################################################################

Pros:
1. The paper overall is well-written and easy to follow.
2. The authors have conducted extensive experiments (quantitative and qualitative) in the paper.

##########################################################################

Cons:
1. [Motivation] The authors worried about training GANs with limited data will lead to memorization of training images (claimed in the abstract). However, the results shown in Figure 1 look like a kind of repetition of patches of the training samples, which conflicts with the motivation.

2. [Motivation] The application of training from a single sample (in the first paragraph on Page 2) is not convincing. Are there any other practical examples to show the practical value of learning from a single sample?

3. [Motivation] The first contribution of learning from a single video is not convincing. Similar to generating a single image by learning from a single image, readers typically expect learning to generate a video by learning from a single video. The authors should provide video results that show temporal coherence. Otherwise, learning from a single video is actually learning from multiple frames.

4. [Approach] Two-branch-like discriminators have been widely explored in GAN. For example, local and global discriminators (Globally and locally consistent image completion. Iizuka et al., TPOG 2017). I think the novelty can be limited.

5. [Approach] Since there is only one real sample during training, I don’t think this two-branch discriminator is able to prevent memorization as claimed on Page 5. I think the effective technique is feature augmentation. Please show the results without feature augmentation and regularization to verify the effectiveness of the discriminator.

6. [Approach] I have trouble understanding the regularization technique in Section 3.2. What does “be more or less equally different from each other” mean? In my view, the regularization technique is actually a maximization of the feature distance of different samples.

7. [Experiments] The results in Figure A is confusing. The results of ”no content branch” also show a worse layout, which is hard to verify the effectiveness of the layout branch and content branch respectively.

8. [Experiments] I suggest ablation studies (by visual comparisons) on the feature augmentation and the diversity regularization, to verify the effectiveness of the proposed model.


**Summary Of The Paper:**

The paper presents a new GAN model that generates new scene compositions from a single training image or a single video clip.  This is achieved by a two-branches (content-layout) discriminator and a diversity regularization technique. The paper also claims as the first work learning from a single video.

**Summary Of The Review:**


Overall, I vote for rejecting. I am especially concerned about the novelty of the two-branch-like discriminator, the effectiveness of the discriminator, and the motivation of learning from a single video. Please address the concerns listed in Cons during the rebuttal period.

---

> ### Author Response · Authors · 2021-11-16
> **Response to Reviewer DXh5**
>
> **DXh5-7:** *“I have trouble understanding the regularization technique in Section 3.2. What does “be more or less equally different from each other” mean? In my view, the regularization technique is actually a maximization of the feature distance of different samples.”*
>
> Our intuition about samples being “similarly different from each other” is linked to our training setups, in which we use only a single image or video for training. Contrary to standard datasets containing images of many different instances or class categories, in these settings all the training images depict the same scene and objects. Therefore, in this case the whole latent space of the generator should model only one semantic scene, in which all the generated samples are highly similar to each other. As the latent space contains no transitions between semantic domains, the perceptual difference between images should be much less correlated to their distance in the latent space. Based on this observation, we design the DR loss to be applied uniformly to all pairs of generated samples, regardless of the distance between their latent codes. As shown in Table 4, this step is extremely beneficial for diversity in the single image setting, improving LPIPS from 0.14 to 0.21.
>
> Our proposed diversity regularization (DR) indeed maximizes the distance between generated samples in the feature space of the generator. Our motivation for using the feature space is based on the observation that generator features are effective at encoding semantic properties of generated images [7, 8]. Thus, by maximizing the distance between samples in the feature space, our DR encourages the generator to produce images with diverse semantic properties, such as varied content and global scene layouts. In our experiments, this solution shows much more effective compared to using the distance in the pixel space, significantly improving LPIPS further from 0.21 to 0.33 in Table 4.
>
> [7] [GAN Dissection: Visualizing and Understanding Generative Adversarial Networks.](https://arxiv.org/abs/1811.10597) Bau et al, ICLR19
> [8] [Repurposing GANs for One-shot Semantic Part Segmentation.](https://arxiv.org/pdf/2103.04379.pdf) Tritrong et al, CVPR21
>
>  $~$
>
> **DXh5-8:** *“The results in Figure A is confusing. The results of ”no content branch” also show a worse layout, which is hard to verify the effectiveness of the layout branch and content branch respectively.”*
>
> Despite having a visual intuition about the branches learning content and layout, we have no explicit intention to achieve their disentangled learning. In our two-branch discriminator, the content and layout representations are obtained from the same intermediate features $F(x)$ via different linear operations, so these features are inherently correlated to each other. This implies that the layout loss $L_{layout}$ also contains information about the image content. This content guidance together with a separate low-level loss $L_{low-level}$ can bias the layout representation, sometimes resulting in less realistic layouts.
>
> Yet, we still observe that the results in Fig. A corelate well with our intuition about the content and layout learning. For example, the model with the content branch (“no layout branch”) learns the content distribution, and thus favors to produce the correct proportions between semantic areas (for example preserving the area for sky, colosseum and ground in the 1st example), sometimes combining them in globally incoherent way (note how colosseum parts can be put on top of each other). In contrast, the model with the layout branch (“no content branch”) tends to produce objects of very different areas (e.g., the area of colosseums and skies varies greatly), while the objects are not distorted into disconnected components. Finally, our final model effectively combines the both branches, generating images with both realistic content and layouts.
>
> We also invite you to have a look at Appendix A.1 (second paragraph) and  Fig. C, where we analyze the feature distances between real images in the content and layout embeddings of the SIV-GAN discriminator, trained on the “bus” and “parkour” videos from Fig. 1 and 5. We find that the feature distances in Fig. C correlate well with our visual intuition, as the layout distances between “parkour” frames are notably higher than for the “bus” video, being in line with the perceptual judgment.
>
>
> $~$
>
>
> In conclusion, we hope that the provided answers clarified the motivation behind our experimental settings and convinced you on the effectiveness and novelty of our technical proposals. We are looking forward to hearing your feedback, and we would be glad to answer any follow-up questions.

---

> ### Author Response · Authors · 2021-11-16
> **Response to Reviewer DXh5**
>
> **DXh5-5:** *“Since there is only one real sample during training, I don’t think this two-branch discriminator is able to prevent memorization as claimed on Page 5. I think the effective technique is feature augmentation.”*
>
> We argue that the two-branch discriminator is a crucial component to prevent the memorization issues in our extremely low data regimes. To illustrate this, we refer to the baseline “No branches” in Table 3, which corresponds to a standard GAN discriminator without our proposed content-layout separation. Note that this baseline is trained with differentiable image augmentation (DA) and uses all other our contributions, such as the Feature Augmentation (FA) and Diversity Regularization (DR). However, such a model does not mitigate overfitting, scoring low in the diversity metrics in Table 3 (e.g., 0.12 and 0.18 Dist. to train). We support this observation in Fig A,B with qualitative results, where the “No branches” baseline suffers from memorization in all examples. Therefore, our proposed content-layout separation with two branches is an effective solution for mitigating memorization, and DA, FA, and DR altogether are not successful without it.
>
> In line with your expectations, our proposed feature augmentation is also an important component to further improve diversity. In our experiments, it significantly boosts the diversity scores, for example improving the LPIPS from 0.27 to 0.33 In the Single Image setting (see Table below). Still, we observe that its effect is not leveraged without the content-layout separation of our two-branch discriminator, as the FA alone does not succeed in enabling synthesis of high diversity (LPIPS of 0.13).
>
> | Two-branch D   | FA | LPIPS $\uparrow$ | Dist. to train $\uparrow$ |
> |---|---|---|---|
> |  |  | 0.11 | 0.09 |
> |  | ✓ | 0.13 | 0.12 |
> | ✓ |  | 0.27 | 0.33 |
> | ✓ | ✓ | **0.33** | **0.37** |
>
> $~$
>
> **DXh5-6:** *“I suggest ablation studies (by visual comparisons) on the feature augmentation and the diversity regularization, to verify the effectiveness of the proposed model.”*
>
> We thank you for the suggestion. We added visual examples for models “No DR” and “No FA” from Table 3 (trained on a single image) to a new Fig. L in the Appendix and provided the analysis in a new section D.5.
>
> We find that the visual results highlight the importance of both DR and FA, corresponding well to the numbers from Table 3. For example, the “No DR” model scores very poorly in diversity metrics (0.04 LPIPS) in Table 3, and the generated images in Fig. L visually exhibit very little diversity, clearly indicating memorization issues. “No FA” model already achieves substantial diversity scores (0.27 LPIPS), but in Fig. L we still observe a rather limited ability to generate high-level variations in content and layout. For example, for an image with waves and surfers, it does not change the number of surfers, while for images with rocks, it typically translates rocks to new locations but does not change their shape. Finally, our full model, with both DR and FA (LPIPS 0.33 in Table 3), enables generating interesting novel scene compositions, in which both content and layouts vary significantly. For instance, such compositions can change the number of surfers and mountains, or modify the shapes of rocks.
>
> Overall, we conclude that all the components of our proposed model are important to achieve high-quality results. Our two-branch discriminator combined with the diversity regularization enables diverse image synthesis, while the feature augmentation introduces the generation of novel interesting combinations of content and layouts.

---

> ### Author Response · Authors · 2021-11-16
> **Response to Reviewer DXh5**
>
> **DXh5-3:** *“The first contribution of learning from a single video is not convincing. Similar to generating a single image by learning from a single image, readers typically expect learning to generate a video by learning from a single video. The authors should provide video results that show temporal coherence. Otherwise, learning from a single video is actually learning from multiple frames.”*
>
> We are sorry for the confusion in our task description. We recall that the focus of our work is to enable unconditional image generation in extremely low data regimes. Therefore, in Sec. 1 (page 2) we introduce the Single Video setting as a task to generate *images*, regarding a video as a collection of *images* with highly correlated content. Generating realistic videos utilizing temporal relations between frames, such as in [5], is a very different task from ours. Thus, we believe that utilizing “learning from a video” and “learning from frames of a video” as synonyms in our image generation setup is meaningful. We thank you for the comment, we plan to adjust the wording in the paper revision to avoid any unclarity on our Single Video setting.
>
> We find the proposed Single Video setting interesting for two reasons. Firstly, from the data collection perspective, it requires recording only one short video, which is affordable in many practical domains. Secondly, this data regime becomes surprisingly challenging for previous one-shot and few-shot GAN models (see Fig. 5 and Table 2), which makes it a new interesting image generation benchmark.
>
>  Note that a dataset obtained from 100 video frames is structurally different from 100 images taken from a standard few-shot dataset (such as the dataset depicting faces of dogs in Fig. E). Particularly, a video provides significantly less variability due to a high correlation between its adjacent frames. With multiple, yet very similar images, the Single Video setting can be intuitively regarded as a mixture of one-shot and classical few-shot regimes. We believe, in the future, our work may foster the development of GANs in similar borderline regimes (small image collections of very little diversity), which can enable the application of GANs to new domains.
>
> [5] [MoCoGAN: Decomposing Motion and Content for Video Generation.](https://arxiv.org/abs/1707.04993) Tulyakov et al, CVPR18.
>
>  $~$
>
> **DXh5-4:** *“Two-branch-like discriminators have been widely explored in GAN. For example, local and global discriminators (Globally and locally consistent image completion. Iizuka et al., TPOG 2017). I think the novelty can be limited.”*
>
> We agree that using several pathways with different objectives in a GAN discriminator is not a novel idea itself. It is indeed a common practice to employ several discriminator heads, for example, to allow outputs of different modalities, such as class labels or image reconstruction. However, we argue that the motivation behind our two-branch discriminator is significantly different from previous works.
>
> In our work, we aim to enable successful GAN learning in extremely low data regimes. Previously, training single-stage GANs on a single image or video was not possible, because a canonical single-stage discriminator was not prevented from memorizing the few training images. By introducing a GAN discriminator with separate content and layout branches, we successfully mitigate the memorization issue, *for the first time* successfully training a single-stage unconditional GAN on a single image and a single video. Moreover, in Appendix C we demonstrate that the proposed two-branch discriminator generalizes to a standard few-shot image generation setting, where it helps to improve the quality, diversity, and data efficiency of the FastGAN model. Therefore, we believe that our proposal brings a solid novelty to the task of unconditional image generation under low data regimes.
>
> Note that the two-branch global and local discriminator of [6] would not be helpful in mitigating the memorization issue. Note that its global part evaluates the realism of the whole image like a standard GAN discriminator. Therefore, similarly to our baseline “No branches” in Table 3 and Fig A, B, such an approach can memorize the whole image at a global scale, and guide the generator away from producing novel scene compositions. The local branch of this discriminator would not mitigate memorization either, despite being trained only on random crops of given training images. In order to match the distributions of fake and real images, such a discriminator would force every random crop of a fake sample to replicate a crop from a training image, thus converging only when fake images fully copy the training examples.
>
> [6] [Globally and locally consistent image completion.](http://iizuka.cs.tsukuba.ac.jp/projects/completion/en/) Lizuka et al, SIGGRAPH 2017

---

> ### Author Response · Authors · 2021-11-16
> **Response to Reviewer DXh5**
>
> Many thanks for your feedback and very valuable comments. We are glad you remarked on the high quality of our experiments and commented positively on our writing. Let us next point-wisely address the cons that you have raised:
>
> $~$
>
> **DXh5-1**: *“The authors worried about training GANs with limited data will lead to memorization of training images (claimed in the abstract). However, the results shown in Figure 1 look like a kind of repetition of patches of the training samples, which conflicts with the motivation.”*
>
> In line with [1], we refer to the *memorization* issue when a GAN converges to reproduce only the copies of training images, being unable to synthesize unseen samples. We note that this is not the case for any example in Fig. 1, where all generated samples demonstrate substantial structural changes compared to the training images. In our extremely low data regimes, the synthesis is inherently constrained by the appearance of objects that are present in the original single image or video. Therefore, a model that does not use any additional data is not expected to create new content, for example, to produce a car of a different model for the first video in Fig. 1. Nevertheless, our model achieves an impressive ability to generate novel scene compositions, for example, to resynthesize the scene without a car or with two cars or change the layout of the mountains in the background (see Fig. 1). Therefore, the results in Fig. 1 do not conflict with our motivation. Finally, as shown in Tables 1,2, our model achieves a higher distance to training data compared to previous one-shot and few-shot GAN models, indicating that SIV-GAN is more successful at mitigating memorization.
>
>  [1] [Theoretical Insights into Memorization in GANs.](https://www.cs.cmu.edu/~vaishnan/papers/GAN_memorization.pdf) Nagarajan et al, NeurIPS18 workshops
>
> $~$
>
> **DXh5-2:** *“The application of training from a single sample (in the first paragraph on Page 2) is not convincing. Are there any other practical examples to show the practical value of learning from a single sample?”*
>
> Diverse image synthesis from very limited data has a potential application to image augmentation. Our model allows us to generate many different and plausible versions of the single sample, which significantly augments the original data but still preserves its semantic context. Therefore, the generated images can be combined with the original image, forming an extended dataset for a practical application aiming to train or fine-tune neural networks on a single image.
>
> As reflected in high *Dist. to train* scores in Tables 1,2, the augmentation produced by SIV-GAN is complementary to standard image augmentation and cannot be achieved with simple geometric transformations, like flipping or rotation. For example, standard augmentation techniques always preserve the relative positions of objects. In contrast, our synthetic augmentations overcome this limitation: as seen in Figure 1, our model can put objects at different locations, swap their positions, or even duplicate or remove some of them. Therefore, we manage to produce image augmentation of higher structural diversity, which can provide additional gains when applied to downstream applications.
>
> Possible examples of downstream applications include one-shot image classification or segmentation, one-shot video object tracking, one-shot object detection [2,3]. Note that in all these examples, at training time a neural network is usually pre-trained on a large dataset, but at test time it is required to adapt to a new domain by using only one image. A common approach to avoid overfitting under such a critical lack of data at test time is to employ image augmentations of the single available image. With our model, we significantly increase the structural diversity of image augmentations, thus bringing a potential for a much stronger reduction of overfitting and further improvement of the network performance.
>
> Moreover, we foresee the application of single image synthesis to very rare examples and dataset outliers. As many practical applications have to deal with largely under-represented classes, sometimes having only one example, e.g., as in [4], our model can serve as an additional source for augmenting such infrequent categories, potentially improving the performance and robustness of algorithms to such outlier cases.
>
> In conclusion, we find that diverse single image synthesis has the potential to improve the performance and data efficiency of practical applications operating under single-image data scenarios.
>
> [2] [Rethinking Few-Shot Image Classification: a Good Embedding Is All You Need?]( https://arxiv.org/abs/2003.11539) Tian et al, Arxiv
> [3] [One-Shot Video Object Segmentation.](https://arxiv.org/abs/1611.05198) Caelles et al, CVPR17
> [4] [LVIS: A Dataset for Large Vocabulary Instance Segmentation](https://arxiv.org/abs/1908.03195) Gupta et al, CVPR19

---

### Official Review · Reviewer_TwJf · 2021-11-02

**Correctness:** 3
**Technical Novelty And Significance:** 2
**Empirical Novelty And Significance:** 2
**Recommendation:** 3
**Confidence:** 5

**Details Of Ethics Concerns:**

The method can generate high-quality images given few data. This method if abused would produce false news and rumors.

**Main Review:**

The proposed method generates good quality results. However, I think the technical novelty is limited and the task of learning from a single video is much simpler than learning from a single image.

1. The method proposes to learn new scene generation from a short video. Although a short video takes similar access effort as an image, it literally contains at least tens or hundreds images. This task is much easier than learning from a single image.

2.The paper didn't use the temporal information in the video, so what's the difference from a video and hundreds of frames? Also, hundreds of images are often enough for image-to-image translation, e.g. [A] uses hundreds of  frames to learn a render-to-realistic GAN.

[A] Yi et al. Audio-driven Talking Face Video Generation with Learning-based Personalized Head Pose. Arxiv.

3. The proposed two-branch discriminator assumed content information is spatial independent, which is strange since spatial distribution also provides much content information. And the method mainly aggregates spatial information and channel information separately, which has also been proposed in previous methods, e.g. [B].

[B] Fu et al. Dual Attention Network for Scene Segmentation. CVPR 2019.

4. The diversity loss is also quite simple, i.e. given two latent codes z1, z2, encourage the generated images G(z1), G(z2) to be different. Shouldn't the distance between two latent codes affect the distance between generated images? In current setting, if z1 and z2 is very close, the loss also encourages the generated images G(z1), G(z2) to be different.

**Summary Of The Paper:**

The paper proposed a new method for generating new scene compositions using a single video. The method proposes a two-branch discriminator, one for content discrimination and the other for layout discrimination. It proposes a diversity loss for generator to encourage images generated from different latent codes to be perceptually different. Experiments show the method generates diverse and high-quality scene images.

**Summary Of The Review:**

In summary, the statement of learning from a single video is basically the same as learning from hundreds of images, and is thus much easier than learning from a single image. The proposed contributions, i.e., the two-branch discriminator and diversity loss, lack technical novelty (see above). So I think the paper is not good enough for acceptance.

---

> ### Author Response · Authors · 2021-11-15
> **Response to Reviewer TwJf**
>
> **TwJf-5:** *“The method mainly aggregates spatial information and channel information separately, which has also been proposed in previous methods, e.g. [B].”*
>
> We agree that the idea of channel and spatial attention has been proposed previously, and in Sec. 3.1 we discuss that our implementation for the content and layout separation is inspired by CBAM [6]. However, we argue that this idea has not been explored as a method to mitigate overfitting and memorization in a GAN discriminator.
>
> Mitigating memorization in GANs is an important problem because it limits the applicability of these models to datasets with few examples. By introducing the content-layout separation to a GAN discriminator, we successfully mitigate the memorization issue, *for the first time* successfully training a single-stage unconditional GAN in such extremely low data regimes as learning from a single image or video. Moreover, in Appendix C we show that our solution generalizes to a standard few-shot image generation setting, showing the benefit of our two-branch discriminator to improve the quality, diversity, and data efficiency of the FastGAN model. Therefore, we believe that our proposal forms a solid contribution to the task of unconditional image generation under low data regimes.
>
> We emphasize that the motivation for the feature restriction in [B] is significantly different from ours. [B] applies the Position and Channel attention modules to enrich the feature representation for the task of semantic segmentation. In this proposal, the attention modules are trained jointly, and the network learns more descriptive semantic features to represent the provided image. Contrary to this approach, as discussed in *TwJf-4*, in our model we strongly rely on the independence of the content and layout discriminator decisions, training them disjointly. Therefore, the solution from [B] would not work for SIV-GAN due to the joint training of modules, which would discourage our generator to produce the combinations of content and layout that were not seen in the training data (see an example in *TwJf-4*).
>
> [6] [CBAM: Convolutional block attention module.](https://arxiv.org/abs/1807.06521) Woo et al, ECCV18
> [B] [Dual Attention Network for Scene Segmentation.](https://arxiv.org/abs/1809.02983) Fu et al, CVPR 2019.
>
> $~$
>
> **TwJf-6:** *“The diversity loss is also quite simple, i.e., given two latent codes encourage the generated images to be different. Shouldn't the distance between two latent codes affect the distance between generated images? In current setting, if z1 and z2 is very close, the loss also encourages the generated images to be different.”*
>
> Our proposed diversity regularization (DR) is indeed simple, yet a very effective technique to boost the diversity of synthesis. The solution to regularize the difference of images based on the distance between their latent codes, e.g., $(G(z1)-G(z2))/(z1-z2)$, is also possible, as was proposed in [7], also known as a Diversity Sensitive loss (DS). Note that [7] studied GAN training on sufficiently large datasets, which is a different setup from ours. In our case of a single image or a single video, the whole latent space of the generator should model only one semantic scene. In this case, all the generated samples belong to the same semantic domain and thus are highly similar to each other. Therefore, compared to training on a large dataset with many different semantic domains, the perceptual difference between generated images should be much less correlated to their distance in the latent space.
>
> In our experiments, this DS loss from [7] leads only to modest diversity (Table 5, shown below). In line with our above analysis, we decided to omit the denominator to make the regularization more aggressive, intentionally encouraging the images to be different even for very close latent codes $z1$ and $z2$. As seen in the Table, this step is extremely beneficial for diversity in the single image setting, improving LPIPS from 0.14 to 0.21. We emphasize that moving the diversity regularization to the generator’s feature space (instead of image space) is also a part of our contribution, which significantly improves the diversity further (LPIPS of 0.33). As concluded from the Table, our proposed DR significantly outperforms other regularizations and is a crucial component of SIV-GAN to achieve high diversity.
>
> | Regularization   | LPIPS $\uparrow$ | Pixel Diversity $\uparrow$ |
> |---|---|---|
> | DS | 0.14 | 0.45 |
> | DR (image space) | 0.21 | 0.52 |
> | DR | **0.33** | **0.66** |
>
> [7] [Diversity-sensitive conditional generative adversarial networks.](https://arxiv.org/abs/1901.09024) Yang et al
>
> $~$
>
> In conclusion, we hope that our answers clarify all the misunderstandings, particularly on the side of the difficulty of the Single Video setting and the novelty of the proposed contributions. We are looking forward to hearing your feedback and hope to have a productive discussion with you.

---

> > ### Comment · Reviewer_TwJf · 2021-12-02
> > **Thanks for the author response**
> >
> > Thanks for the author response.
> >
> > After reading the author response, I acknowledge the difficulty of unconditional image generation from 100 frames without pre-training; and understand the content-separation branch design (without the independence assumption).
> >
> > However,
> > 1)  I still think the difference between a video and 100 images is much smaller than the difference between a video and a single image. The author said: "the diversity of images in such datasets is different: videos usually have much lower variability due to a high correlation between their adjacent frames", if this is the case, the variability of a single image is near 0 so single image is a lot more challenging.
> >
> > 2) For channel and spatial separation, though it is designed for a new motivation (mitigating memorization of GAN discriminator), the methodology design is similar to existing works like [B], which limits the original technical contribution.
> >
> > 3) The method "intentionally encourage the images to be different even for very close latent codes z1 and z2", which helps achieve high diversity, but ignores the distance in the latent space.
> >
> > So I decided to retain my rating.

---

> > > ### Author Response · Authors · 2021-12-03
> > > **Reply to TwJf**
> > >
> > > Thanks for your feedback on our answers. We, unfortunately, could not agree with your points. Especially, we could not understand the reasons for which the points 1 and 3 are considered as shortcomings of our work.
> > >
> > > 1. We did not claim that one setting is harder than another. We just argued that the video setting is very challenging. While there exist successful models for single images (e.g., SinGAN [1]), none of the baselines in Sec. 4 successfully learns from a single video. In this regime, our model for the first time enables the synthesis of high quality and diversity. We are glad you acknowledged the difficulty of this task.
> > >
> > > 2. We do not agree that the methods that inspired us from other research fields should be the reason for reduced novelty. As outlined in *TwJf-3*, previous methods would not solve our task, being unhelpful to mitigate the problem of memorization. The combination of our proposed content-layout D, DR, and FA enables learning in a new challenging setting, also improving image quality and diversity across different data regimes. Thus, we believe that our proposals bring a solid novelty to the task of unconditional image generation under low data regimes.
> > >
> > > 3. We empirically demonstrated that using the distance in the feature space is not beneficial in our setting and provided the explanation for this observation (see *TwJf-6*). We therefore could not understand why this insight is regarded as a disadvantage of the approach.

---

> ### Author Response · Authors · 2021-11-15
> **Response to Reviewer TwJf**
>
> **TwJf-3:** *“Also, hundreds of images are often enough for image-to-image translation, e.g. [A] uses hundreds of frames to learn a render-to-realistic GAN.”*
>
> As you have mentioned, the paper [A] uses a GAN for image-to-image translation. Please note that the focus of our work is significantly different, as we study *unconditional image generation* in extremely low data regimes, for which we assume that no additional data is available for pre-training.  In contrast, training image-to-image models usually requires large collections of images, in both the source and target domains. Similarly, the approach in [A] is pre-trained with a large collection of videos, as described in their Sec. 3 (“we propose a novel memory-augmented GAN module (Section 3.4) that was also trained by the LRW video dataset”). Therefore, their approach has access to *hundreds of thousands of short videos* from the LRW dataset [5] during training. Thus, we observe a significant difference in our settings, as we aim to train our model *only on one short video*, refraining from using any pre-training data. Note that using pre-training is not always an acceptable solution, as a possible semantic incompatibility between the source and target domains can be harmful to performance [4].
>
> [A] [Audio-driven Talking Face Video Generation with Learning-based Personalized Head Pose.](https://arxiv.org/abs/2002.10137) Yi et al, Arxiv
> [4] [Differentiable augmentation for data-efficient gan training.](https://arxiv.org/abs/2006.10738) Zhao et al, NeurIPS20
> [5] [LRW dataset](https://www.robots.ox.ac.uk/~vgg/data/lip_reading/lrw1.html)
>
> $~$
>
> **TwJf-4:** *“The proposed two-branch discriminator assumed content information is spatial independent, which is strange since spatial distribution also provides much content information.”*
>
> We regret to hear that you found the motivation behind separating the content and layout features in our discriminator unclear. We kindly point out that contrary to your comment, we do *not* assume independence of the content and layout features, as they are indeed obtained from the same feature tensor $F(x)$ via two simple squeezing operations. Instead, our approach assumes only the independence of the content and layout discriminator decisions (the content and layout losses $L_{D_{content}}$ and $L_{D_{layout}}$).
>
> We would like to illustrate our idea again with a simple new example. Let us create an artificial transformation of the image with air balloons from Fig. 1 (3rd row), in which we exchange the positions of the yellow and dark blue balloons. Note that such transformation would not change the feature representation of the image in both content and layout embeddings: the content representation is invariant to changes in locations of objects, while the layout map is insensitive to the changes in object colors. Therefore, such a transformed image seems real to both the content and layout branches. This corresponds to our motivation because now the generator is encouraged to produce such a novel scene composition. Note that this would not be possible without the content-layout separation, as the discriminator without the branches can learn the colors and locations of objects jointly, which would penalize generating balloons in different locations. Finally, as we show in Sec. 4.2, our idea to separate the layout and content discriminator decisions is indeed helpful to mitigate the discriminator memorization, being a crucial component to achieve high diversity of synthesis. We invite you to have a look at Sec. A in the Appendix, where we provide additional analysis of our proposed two-branch discriminator.

---

> ### Author Response · Authors · 2021-11-15
> **Response to Reviewer TwJf**
>
> We thank you for your time reviewing our paper. Based on your review, we believe that you have unfortunately misunderstood our contributions. We are sorry if this is in some part caused by unclarity in our presentation. In what follows we will do our best to provide clarification on the details of our work and answer all your questions.
>
> $~$
>
> **TwJf-1:** *“The method proposes to learn new scene generation from a short video. Although a short video takes similar access effort as an image, it literally contains at least tens or hundreds of images. This task is much easier than learning from a single image.”*
>
> We respectfully disagree that the Single Video setting is easier than learning from a single image. Please note that this setting is unconditional, and there is no access to any other data for pre-training. As we demonstrate in Table 2 and Fig. 5, state-of-the-art models for one-shot and few-shot image synthesis, like SinGAN [1] and FastGAN [2], are not able to successfully learn in this setting. For instance, in Fig. 5 SinGAN does not achieve good image quality, distorting objects and producing unrealistic scene layouts. In Table 2 this corresponds to very high SIFID scores. The few-shot model FastGAN, on the other hand, cannot achieve good diversity, reproducing only the frames from the training videos (note very low Dist. to train. In Table 2). Therefore, our proposed setting is difficult for both one-shot and few-shot image generation models, so it introduces novel challenges compared to previous one-shot and few-shot experimental setups.
>
> We note that increasing the number of frames from 1 to 100 does not imply an easier setting. Training unconditional GANs from scratch in low data regimes, such as from 100 images, is well known to be difficult due to the overfitting that occurs in discriminators [2, 3]. Current state-of-the-art data-efficient unconditional GANs, such as FastGAN [2], were shown to successfully learn on datasets consisting of several hundreds of images. We invite you to have a look at our Appendix Sec. C, particularly at Fig. E, where we show that FastGAN learns successfully from a dataset with 389 faces of dogs, but fails when the number of images is decreased to 98. Therefore, learning from 60-100 video frames is a challenging few-shot setup due to the small number of images available for training.
>
> [1] [SinGAN: Learning a Generative Model from a Single Natural Image.](https://arxiv.org/abs/1905.01164) Shaham et al, ICCV19
> [2] [Towards Faster and Stabilized GAN Training for High-fidelity Few-shot Image Synthesis.](https://arxiv.org/abs/2101.04775) Liu et al, ICLR21
> [3] [Training Generative Adversarial Networks with Limited Data.](https://arxiv.org/abs/2006.06676) Karras et al, NeurIPS20
>
> $~$
>
> **TwJf-2:** *“The paper didn't use the temporal information in the video, so what's the difference from a video and hundreds of frames?”*
>
> As we describe in Sec.1 (page 2), our work is focused on unconditional image generation. We, therefore, introduce the Single Video setting as a novel challenging regime for image generation, and we don’t aim at generating realistic videos. Generating realistic videos utilizing temporal relations between frames, such as in [4], is a very different task from ours. Thus, we believe that utilizing “learning from a video” and “learning from frames of a video” as synonyms in our image generation setup is meaningful.  We are sorry if our wording caused your confusion about our experimental setup.
>
> Learning from 100 frames of a single video in our formulation is technically indeed similar to training on any other dataset containing 100 images. However, the difficulty of learning from a single video is not the same as learning from datasets commonly used for few-shot generation, such as a dataset depicting faces of dogs (see Fig. E). This happens because the diversity of images in such datasets is different: videos usually have much lower variability due to a high correlation between their adjacent frames. To measure this effect quantitatively, we compute the LPIPS diversity for the few-shot “Face-dog” dataset from Fig. E and for the single video of a car following a road from Fig 1. We obtain the LPIPS of 0.63 and 0.38 respectively, meaning that a single video provides a significantly less diverse set of images compared to a standard few-shot dataset of the same size. As seen from Fig. 5, such a lack of diversity challenges a few-shot model FastGAN, which is not able to overcome memorization issues in this scenario.
>
> [4] [MoCoGAN: Decomposing Motion and Content for Video Generation.](https://arxiv.org/abs/1707.04993) Tulyakov et al, CVPR18.

---

### Official Review · Reviewer_inBU · 2021-11-03

**Correctness:** 3
**Technical Novelty And Significance:** 2
**Empirical Novelty And Significance:** 2
**Recommendation:** 5
**Confidence:** 4

**Main Review:**

This paper improve the sinGAN on image generation using very few training images. Technically speaking, the proposed solution is interesting, but the technical contribution is not super significant. It is not new to disentangle the image into content and layout parts by restricting the feature dimension -- deep vector for content without spatial information and shallow map for layout without semantic information, similar to the papers such as Swapping Autoencoder. The results look better than sinGAN, but still suffer from large duplicated content and distortion artifacts, but that's understandable for such an extremely challenging problem.

For the technical details, the paper is written well and easy to understand. I just have a few small questions and comments:
1. In feature augmentation, what exactly does "For the single image setting this is done by mixing the features of two different augmentations of the original image"? What is the different augmentation here?
2. In feature augmentation again, I am a bit concerned about the proposed augmentation. Take layout as an example, to crop a rectangle of feature to another image does not make a reasonable real image in my understanding, which is equal to crop a random patch from one frame and paste to another frame, the composted image will look strange. I am either not sure about the content augmentation. A discussion could help understand the intuition.
3. No visual comparisons for many ablation analysis, including DR and FA.
4. Would it make sense to add another reconstruction loss by combining content and layout features through another decoder to form an autoencoder? It might help enforce feature completeness and make training easier? But I'm not sure.
5. Are those editing applications in sinGAN (Fig.12) supported?

**Summary Of The Paper:**

This paper proposes an approach to train an image generator using only a short video clip or a single image. The overall framework follows the design of sinGAN that has an unconditional generator that generates new images from random noises, and the generated images and the input real images/frames are fed to discriminators to enforce similar distributions. The key contribution is to disentangle the image into content and layout features, and use different discriminators, which is claimed to reduce the memorization issue.

**Summary Of The Review:**

Overall I like this paper, which is solid, complete, with sufficient experiments and evaluations. I am okay with acceptance, but I am a bit concerned that the contribution might not be significant enough for acceptance, since it is essentially the sinGAN setting plus content/layout separation.

---

> ### Author Response · Authors · 2021-11-15
> **Response to Reviewer inBU**
>
> **inBU-7**: *“Would it make sense to add another reconstruction loss by combining content and layout features through another decoder to form an autoencoder? It might help enforce feature completeness and make training easier? But I'm not sure.”*
>
> Thank you for the interesting idea! Indeed, adding a reconstruction loss to the discriminator objective has recently been shown as an effective self-supervision technique for few-shot image generation [2]. Similarly, we have tried to regularize our $D_{low\text{-}level}$ discriminator with a reconstruction loss or a via auxiliary rotation loss from [4]. We collect the results in the table below for the single image setting. We, unfortunately, observed that self-supervision is not helpful for GAN training in our extremely low data regime. We observed that it didn’t improve the quality of images (note same or higher SIFID), while the diversity among images became lower (lower LPIPS). We hypothesize that with just 1 image the reconstruction task of an extra decoder or rotation predictor becomes trivial, which in turn allows the discriminator to memorize the training frames easier.
>
> | Model | SIFID $\downarrow$ | LPIPS $\uparrow$ |
> |---|---|---|
> | SIV-GAN | **0.08** | **0.33** |
> | + reconstruction | **0.08** | 0.31 |
> | + rotation | 0.12 | 0.27 |
>
> For the idea about learning a decoder from combined content and layout features, as in the Swapping Autoencoder, we note that this solution would not be in line with our motivation. In order to mitigate overfitting and memorization, we actually want $D_{content}$ and $D_{layout}$ to learn separately from each other by minimizing $L_{content}$ and $L_{layout}$ in a disjoint manner. As also discussed in answer *inBU-2*, our discriminator mitigates memorization better when the original image cannot be reconstructed from the reduced content and layout representations.  In contrast, a reconstruction loss after joining the content and layout features would force these two embeddings to be mutually dependent, in a way that the given image can be reconstructed from them. Such limitation would constrain the ability of our feature augmentation, for example, by penalizing the layout representation containing objects that were removed from the content features.
>
> [4] [Self-Supervised GANs via Auxiliary Rotation Loss](https://arxiv.org/abs/1811.11212), Chen et al, CVPR21
>
> $~$
>
> **inBU-8:** *“Are those editing applications in sinGAN (Fig.12) supported?”*
>
> Similarly to our response *inBU-1*, we would elaborate that the goal of our work is not solely focused on single image generation, so we do not follow the setting of SinGAN fully. Instead, we aim to enable successful learning of unconditional GANs in extremely low data regimes, which in our definition also includes learning from multiple images. Our model in the current form is thus conditioned only on input noise and does not support any other source of conditional information, which is needed for image editing. As other unconditional GAN models, SIV-GAN supports image editing via latent space exploration [5]. In such a pipeline, an image of interested can be embedded into the generator’s latent space [6], and then the manipulation can be performed via modifying the latent vector. We consider this as an interesting direction for future work.
>
> It is noteworthy that the scope of our applications is different to image editing. Our model enables the learning of GANs in extremely low data regimes, which suggests a potential to generate useful data augmentation in restricted image domains, where data collection remains challenging or expensive. Note that our model is well suited for data augmentation thanks to its ability to generate unseen examples, that are not achievable with standard data augmentation techniques (see the high Dist. to train. scores in Tables 1, 2).
>
> [5] [Image2StyleGAN: How to Embed Images Into the StyleGAN Latent Space?](https://arxiv.org/abs/1904.03189) Abdal et al, ICCV19
> [6] [Unsupervised Discovery of Interpretable Directions in the GAN Latent Space](https://arxiv.org/abs/2002.03754), Voynov et al, ICML20
>
> $~$
>
> We hope that our answers will be helpful for you to clarify the misunderstood points, and to re-assess the significance of our work. We are looking forward to hearing your feedback on our response.

---

> ### Author Response · Authors · 2021-11-15
> **Response to Reviewer inBU**
>
> **inBU-5:** *“In feature augmentation again, I am a bit concerned about the proposed augmentation. Take layout as an example, to crop a rectangle of feature to another image does not make a reasonable real image in my understanding, which is equal to crop a random patch from one frame and paste to another frame, the composted image will look strange. I am either not sure about the content augmentation. A discussion could help understand the intuition.”*
>
> We find your assumption about leakage of the feature augmentations more than reasonable. To illustrate the intuition of why this is not happening, let us compare two different strategies for executing rectangular copy-pasting augmentations. In the first strategy, we apply such transformations on a batch of real samples in the *image space*, before the first discriminator layer. In the second strategy, the augmentation is applied to real images in their *content and layout feature embeddings*.
>
> In our experiments, the first strategy leads to a leakage of copy-pasting augmentations into generated images. Particularly, we observed unnatural rectangular-shaped artifacts in generated samples, for example, small patches of sky textures sometimes appeared on top of foreground objects. Such behavior is generally expected, because augmentation at the image level shifts the distribution of real images, and thus the discriminator learns to recognize such artifacts as real.
>
> In contrast, with the augmentation in the content and layout feature embeddings we do not observe such leakage. Note that as shown in Eq. 1, our discriminator loss is computed after each layer, including the layers of $D_{low-level}$. As the Feature Augmentation is applied only after $D_{low-level}$, $D_{low-level}$ never sees the copy-pasting augmentations. Thus, the low-level loss $L_{D_{low\text{-}level}}$ has a chance to guide the generator away from producing unnatural transitions between semantic areas. Therefore, only globally coherent scene compositions are composed, corresponding to duplication or removal of objects (without cutting), as well as meaningful changes of background layouts. Another reason for the reduced leakage is the difference in strategies for the content and layout augmentations. At each step, these augmentations are applied independently. It can therefore happen that a layout representation contains the objects that were removed from the content representation, leading the gradients from $L_{D_{content}}$ and $L_{D_{layout}}$ to guide the generator in different directions, thus reducing the possible augmentation leakage.
>
> $~$
>
> **inBU-6:** *“No visual comparisons for many ablation analysis, including DR and FA.”*
>
> We thank you for the suggestion. We added visual examples for models “No DR” and “No FA” from Table 3 (trained on a single image) to a new Fig. L in the Appendix and provided the analysis in a new section D.5.
>
> We find that the visual results correspond well to the diversity scores numbers from Table 3. For example, the “No DR” model scores very poorly in diversity metrics (0.04 LPIPS) in Table 3, and the generated images in Fig. L visually exhibit very little diversity, clearly indicating memorization issues. “No FA” model already achieves substantial diversity scores (0.27 LPIPS), but in Fig. L we still observe a rather limited ability to generate high-level variations in content and layout. For example, for an image with waves and surfers, it does not change the number of surfers, while for images with rocks, it typically translates rocks to new locations but does not change their shape. Finally, our full model, with both DR and FA (LPIPS 0.33 in Table 3), enables generating interesting novel scene compositions, in which both content and layouts vary significantly. For instance, such compositions can change the number of surfers and mountains, or modify the shapes of rocks.

---

> ### Author Response · Authors · 2021-11-15
> **Response to Reviewer inBU**
>
> **inBU-2:** *“It is not new to disentangle the image into content and layout parts by restricting the feature dimension -- deep vector for content without spatial information and shallow map for layout without semantic information, similar to the papers such as Swapping Autoencoder.”*
>
> We acknowledge that the restriction of the feature dimension is not a novel idea itself and in Sec. 3.1 we discuss that our implementation for the content and layout separation is inspired by CBAM [1]. However, we argue that this idea has not yet been explored as a method to mitigate memorization in a GAN discriminator. As also discussed in answer *inBU-1*, by introducing the content and layout separation to a GAN discriminator, we *for the first time* manage to successfully train a single-stage unconditional GAN in extremely limited data regimes, such as learning from a single image or video.
>
> We emphasize that the motivation for the feature restriction used in the Swapping Autoencoder [3] is significantly different from ours. [3] studies the task of image manipulation, and their solution aims to disentangle the texture and structure representations that are used as conditioning information for the generator. Therefore, the generator in [3] should be able to reconstruct the original image from these two representations. In contrast, we focus on unconditional image generation, apply the separation in the discriminator, and we have a different goal of memorization mitigation. Importantly, in our setting, we don’t need to be able to reconstruct the original feature $F(x)$ from its layout and content representations. Moreover, we apply different feature augmentations (FA) on the content and layout representations independently from each other, which makes such reconstruction even impossible. As discussed in Sec. 4.2, both the content-layout separation and FA are beneficial in our setting to mitigate overfitting and achieve higher diversity. This way, we benefit from complicating the reconstruction of the original image via the content-layout separation, contrary to [3], which strongly relies on the reconstruction ability of the decoder from separated features.
>
> [3] [Swapping Autoencoder for Deep Image Manipulation](https://arxiv.org/abs/2007.00653), Park et al, NeurIPS20
>
> $~$
>
> **inBU-3:** *“The results look better than sinGAN, but still suffer from large duplicated content and distortion artifacts, but that's understandable for such an extremely challenging problem.”*
>
> We are glad that you acknowledge the challenge behind training GANs from single data instances, when a model has access only to a single image or a video. Inherently, under such a limited scenario, the synthesis of our model is constrained by the appearance of the objects present in the original sample. For example, for the video with a gray car following a road from Fig. 1, we don’t expect that SIV-GAN produces a car of a different color. Therefore, the duplication of content is generally expected due to a limited set of objects in the training data. Nevertheless, even trained with such limited data, SIV-GAN achieves an impressive ability to generate novel scene compositions. For example, it can resynthesize the scene without a car or with two cars, or change the layout of the mountains in the background (see Fig. 1). These results indicate that our model does not simply copy the training examples, supporting our claim on the successful mitigation of the memorization issue.
>
> $~$
>
> **inBU-4:** *“In feature augmentation, what exactly does "For the single image setting this is done by mixing the features of two different augmentations of the original image"? What is the different augmentation here?”*
>
> As we describe in the training details (see Sec. 3.3), we also use differentiable image augmentation (DA) for both real and fake images before feeding them to the discriminator. For the single image setting, this means that inside a real batch we have the original image that is augmented in different ways, using the standard transformations like translation, cropping, rotation, and horizontal flipping. Our FA thus mixes the content and layout features extracted from two different augmentations (for example, horizontal translation and rotation by a small angle) of the same image.

---

> ### Author Response · Authors · 2021-11-15
> **Response to Reviewer inBU**
>
> Thank you very much for your detailed feedback and valuable questions. We are glad that you remarked on the high quality of our experimental evaluation and found our technical proposal interesting. Let us next share our perspective on the questions and concerns you have raised:
>
> $~$
>
> **inBU-1:** *“I am a bit concerned that the contribution might not be significant enough, since it is essentially the sinGAN setting plus content/layout separation.”*
>
> We respectfully disagree that our contribution can be summarized as introducing the content-layout separation to the setting proposed by SinGAN [1]. The overall motivation for our technical proposal is indeed similar to theirs, as we also aim to mitigate GAN memorization in very low data regimes. However, **we argue that the technical solution behind our approach is drastically different from the one of SinGAN**.
>
> SinGAN proposed a patch-based multi-scale GAN, in which different GAN levels are trained consequently in several stages. This solution mitigates memorization of a given image, as the employed patch-GANs at different levels have small receptive fields and limited capacity. Although such a model can produce different images, we identify the disadvantages of patch-based multi-stage approaches (see Fig. 3).
>
> In our work, we introduce a completely different solution to mitigate the memorization issue. Most importantly, we refrain from the multi-stage patch-based approach and introduce a GAN model that learns in a *single stage*. Previously, training single-stage GANs on a single image was not possible, because a canonical single-stage discriminator was not prevented from memorizing the original image (as shown in Fig. 4 and H, a few-shot model FastGAN [2] memorizes the given image). We solve this issue with our proposed contributions: the content-layout separation, a new feature augmentation, and a diversity regularization. With the proposed modifications, we not only enable single image learning with a single-stage GAN, but also achieve higher quality and diversity compared to SinGAN (see Table 1), avoiding object distortions and incoherencies in global layouts (see Fig. 4,G,H).
>
> Besides notable architectural differences, **our model offers a variety of data regimes to which SinGAN cannot be applied to**. Designed to learn from a single image, SinGAN cannot successfully learn from multiple images because of its patch-based multi-stage discriminator, which is not designed to model high-level semantic scene properties. For multi-image datasets, such architecture produces meaningless patch permutations without global coherency. We refer to Fig. 5 and Table 2, showing poor performance of SinGAN when trained on multiple frames of a video. [*] reported similar problems when training SinGAN on a dataset consisting of only two images.
>
> In contrast, a one-stage structure of our model makes it more similar to standard GAN approaches. As we show in Fig 1, 5, our model learns not only from a *single image*, but also successfully learns from *multiple* video frames (we use 60-100 frames in the videos). We also invite you to have a look at Appendix C, particularly Fig. E, F, where we demonstrate the ability of our discriminator to be applied in a classical few-shot image generation setting, such as learning from datasets containing 25, 50, 100, 160 or 389 images.
>
> With the same model configuration, SIV-GAN successfully learns from one or multiple images. To the best of our knowledge, we are the first to design such a unified approach, covering both *one-shot* and *few-shot image* synthesis settings. As we show in the experiments, this helps to extend the applicability of GANs to a fundamentally new data regime – training from frames of a single video. This setting can be intuitively regarded as a mixture of one-shot and classical few-shot regimes - it captures only one scene, but provides multiple (yet very similar) images. As we show in the experiments (see Fig. 5), diverse and high-quality synthesis in the Single Video setting is not achievable for the previous one-shot and few-shot GAN approaches.
>
> In the future, our work may foster the development of GANs on other extremely low data settings that are in between one-shot and classical few-shot training regimes, which can enable the application of GANs to new domains. That being said, we would kindly ask you to reevaluate the significance of our contribution. We hope that the above explanations can now convince you that our work brings a solid novelty, especially with respect to significantly extending upon prior single image generation works (e.g. SinGAN).
>
> [*] [Blog post](https://deepganteam.medium.com/making-singan-double-8568490b572e)
> [1] [SinGAN: Learning a Generative Model from a Single Natural Image.](https://arxiv.org/abs/1905.01164) Shaham et al, ICCV19
> [2] [Towards Faster and Stabilized GAN Training for High-fidelity Few-shot Image Synthesis](https://arxiv.org/abs/2101.04775), Liu et al, ICLR21

---

### Author Response · Authors · 2021-11-19
**General Response**

Dear all reviewers,

We truly appreciate your time and efforts in reviewing our paper. We are glad that you acknowledged the high quality of our results (**TwJf**, **LNdN**), the extensiveness of our experimental evaluation (**inBU**, **DXh5**, **LNdN**), the overall quality of writing (**inBU**, **DXh5**), as well as that you found the proposed solutions interesting and intuitive (**inBU**, **LNdN**).

In the individual responses, we made significant efforts to provide comprehensive answers to all the questions and concerns you have raised:
- As suggested by reviewers **inBU** and **DXh5**, we extended our qualitative analysis with a new section D.5 and Figure L, where we discuss the visual effect of the Diversity Regularization and Feature Augmentation on the synthesis diversity. We observe that the effect of the components corresponds well to the numbers in Table 3 and to our visual intuition behind these technical proposals.
- Following the advice from reviewer **LNdN**, we conducted a human perception study to compare our synthesis quality to previous single-image approaches. The results of the study confirm the higher quality of image generation achieved by our model, being in line with the quantitative and qualitative comparisons in Sec. 4.1.
- Finally, we provided extensive discussions on all the other points from your reviews. We hope they convinced you of the merits of our contributions, particularly on the side of novelty, the difficulty of the studied settings, and the capability of our model standing out from both previous one-shot and few-shot GAN models.

We are gratefully waiting for your feedback and sincerely hope to have further productive discussions with you. We are confident that our responses should have cleared up all the misunderstood points. Should there something remain unclear, we would be more than happy to answer any additional questions and provide more information.

---

### Author Response · Authors · 2021-11-29
**Summary of the Rebuttal**

Dear AC,

As we are approaching the end of the discussion phase, we would take an opportunity to share our comments on the feedback posted by the reviewers.

We have unfortunately not received the feedback from reviewers **inBU**, **TwJf** on our rebuttal. We believe that we provided sufficient answers to address all the concerns raised in their reviews.
We thank the reviewers **DXh5**, **LNdN** for answering our points. Based on the answers, we think that our author responses have unfortunately not been taken into enough consideration:

- *DXh5* commented on two points: novelty and motivation of the single video setting. We find that the comment on novelty misses a significant part of our answer *DXh5-4*, where we highlight the novelty of our content-layout discriminator for the task of unconditional synthesis in low-data regimes. The second comment on the single video setting does not acknowledge the major part of answer *DXh5-3*, highlighting the challenge of the introduced setting and its potential impact on the development of unconditional GAN models in extremely low data regimes.
- *LNDN* is unconvinced by our model not supporting conditional synthesis. In *LNDN-1* we provided a clarification that we have a different goal of unconditional synthesis in low data regimes, and that we have significantly different motivation compared to previous single-image GAN models. Unfortunately, these points seem to have not been considered in the feedback on our response.

Kind regards,
Authors of Paper 2592

---

### Decision · Program_Chairs · 2022-01-20

**Decision:**

Reject

**Comment:**

The authors consider the problem of unconditional image generation in the low-data regime, such as learning from frames of a single video or even from a single image. The main idea is to apply GANs with a specific two-branch discriminator architecture such that the content features and layout features are handled independently. Secondly, to improve the variability of generated images the authors apply diversity regularization. The authors show that the proposed model is able to, to a certain extent, generate diverse high quality samples.

The paper is well-written, the authors described their method and the evaluation protocol thoroughly and clearly. The reviewers felt that this submission was borderline, with questionable novelty and significance. In an extensive rebuttal and discussion phase the authors addressed several raised challenges and improved their paper. However, two points remain:
- **Technical novelty**: content and layout separation as well as diversity regularisation previously appeared in many contexts and papers.
- **Motivation and practicality**: One of the main arguments for the utility of the proposed method is to use it for data augmentation. While it may indeed result in content-based augmentations, it nevertheless necessitates training of a GAN for every single image, which is severely limiting in practice.

After reading the manuscript, reviews, and the rebuttals, my view is that the paper is below the acceptance bar and I agree with the points on novelty and significance. In particular, the main application to data augmentation seems to be "unexciting" and the proposed method impractical. At the same time the proposed method is a combination of already known techniques, albeit in a different setting. I suggest the authors condense the arguments in the extensive rebuttal to improve the points raised above and resubmit.